# PROBABILISTIC STABILITY OF STOCHASTIC GRADIENT DESCENT

## ABSTRACT

Characterizing and understanding the stability of Stochastic Gradient Descent (SGD) remains an open problem in deep learning. A common method is to utilize the convergence of statistical moments, esp. the variance, of the parameters to quantify the stability. We revisit the definition of stability for SGD and propose using the *convergence in probability* condition to define the *probabilistic stability* of SGD. The probabilistic stability shows that alongside local minima, low-capacity saddle points are also potential and likely solutions that SGD converge to after training. We show that only through the lens of probabilistic stability does SGD exhibit rich and practically relevant phases of learning, such as the phases of the complete loss of stability, incorrect learning where the model captures incorrect data correlation, convergence to low-rank saddles, and correct learning where the model captures the correct correlation. These phase boundaries are precisely quantified by the Lyapunov exponents of the dynamics. The obtained phase diagrams imply that SGD prefers low-rank saddles in a neural network when the underlying gradient is noisy.

## 1 INTRODUCTION

Stochastic gradient descent (SGD) is the primary workhorse for optimizing neural networks. As such, an important problem in deep learning theory is to characterize and understand how SGD selects the solution of a deep learning model, which often exhibits remarkable generalization capability. At the heart of this problem lies the *stability* of SGD because the models trained with SGD stay close to the attractive solutions where the dynamics is stable and moves away from unstable ones. Solving this problem thus hinges on having a good definition of the stability of SGD. The stability of SGD is often defined as a function of the variance of the model's parameters or gradients during training. The hidden assumption behind this mainstream idea is that when the variance diverges, the training becomes unstable (Wu et al., 2018; Zhu et al., 2018; Liu et al., 2020; 2021; Ziyin et al., 2022b). In some sense, the idea that the variance of the parameters matters the most is also an underlying assumption in the deep learning optimization literature, where the utmost important quantity is how fast the variance and the expected distance of the parameters decay to zero (Vaswani et al., 2019; Gower et al., 2019). We revisit this perspective and show that a variance-based notion of stability is insufficient to understand the empirically observed stability of training of SGD. In fact, we demonstrate a lot of natural learning settings where the variance of SGD diverges, yet the model still converges with high probability.

In this work, we study the *convergence in probability* condition to understand the stability of SGD. We then show that this stability condition can be quantified with the Lyapunov exponent (Lyapunov, 1992) of the optimization dynamics of SGD, a quantity deeply rooted in the study of dynamical systems and has been well understood in physics and control theory (Eckmann & Ruelle, 1985; Diaconis & Freedman, 1999). Importantly, we apply this notion of stability to understand the attractivity and repulsiveness of saddle points in neural networks. This focus of ours differentiates our work from the previous works that also apply Lyapunov-type conditions to study local minima (Gurbuzbalaban et al., 2021; Hodgkinson & Mahoney, 2021). The main contribution of this work is to propose a new notion of stability that sheds light on how SGD can select saddle points as solutions and discover multiple deep-learning phenomena that can only be understood in terms of this notion. Perhaps the most important implication of our theory is the characterization of the highly nontrivial and practically important phase diagram of SGD of a neural networks close to saddle points.

## 2 PROBABILISTIC STABILITY

In this section, we introduce probabilistic stability, a rather general concept that appears in many scenarios in control theory (Khas' minskii, 1967; Eckmann & Ruelle, 1985; Teel et al., 2014). The definition of probabilistic stability relies on the notion of convergence in probability. We use $\|\cdot\|_p$ to denote the $p$−norm of a matrix or vector and $\|\cdot\|$ to denote the case where $p = 2$. The notation $\to_p$ indicates convergence in probability.

**Definition 1.** *A sequence (of random variables) $\{\theta_t\}_t^\infty$ is probabilistically stable at a constant vector $\theta^*$ if $\theta_t \to_p \theta^*$.*

Here, the notation $\to_p$ denotes convergence in probability. A sequence $\theta_t$ converges in probability to $\theta^*$ if $\lim_{t\to\infty} \mathbb{P}(\|\theta_t - \theta^*\| > \epsilon) = 0$ for any $\epsilon > 0$.

We will see that this notion of stability is especially suitable for studying the stability of the saddle points in SGD, be it a local minimum or a saddle point. In contrast, the popular type of stability is based on the convergence of statistic moments, which we also define below.

**Definition 2.** *A sequence $\{\theta_t\}_t^\infty$ is $L_p$-norm stable at $\theta^*$ if $\lim_{t\to\infty} \mathbb{E}\|\theta_t - \theta^*\|_p^p \to 0$.*[1]

For deep learning, the sequence of $\theta_t$ is the model parameters obtained by the iterations of the SGD algorithm for a neural network. For dynamical systems in general and deep learning specifically, it is impossible to analyze the convergence behavior of the dynamics starting from an arbitrary initial condition. Therefore, we have to restrict ourselves to the neighborhood of a given stationary point and consider the linearized dynamics around it. The main application of moment-based stabilities is to analyze the attractivity of local minima of neural networks, where a key insight is that flatter minima are favored over sharper ones at a large learning rate (Wu et al., 2018; Wu & Su, 2023; Xie et al., 2020). In contrast, the focus of our work is on the attractivity of saddle points.

The type of dynamics we consider in this work are all of the following form:

$$\theta_{t+1} = \theta_t - \lambda \hat{H}(x_t)(\theta_t - \theta^*) + O(\|\theta_t - \theta^*\|^2), \tag{1}$$

where $\lambda$ is the learning rate, $\theta^*$ is the critical point under consideration, $\hat{H}(x_t)$ is a random symmetric matrix that is a function of the random variables $x_t$, which can stand for both a single data point or a minibatch of data points.[2] In this work, $\theta^*$ is said to be a local minimum if $H := \mathbb{E}_x[\hat{H}(x)]$ is positive semi-definite (PSD) and is a saddle point otherwise.

Note that $\hat{H}(x)$ does not have to be tied to a single data point but can also be the Hessian of a minibatch of data points. Essentially, it is the data distribution of $\hat{H}$ that matters. When we have the same data distribution but a different batch size $S$, the distribution of $\hat{H}$ is different. This linearized dynamics is especially suitable for studying two types of stationary points that appear in modern deep learning: (1) interpolation minima and (2) symmetry-induced saddle points. Let us first consider the stability of the interpolation minimum.

A minimum is said to be an interpolation minimum if the loss function reaches zero for all data points. This means that close to such a minimum, the per-sample loss functions all have vanishing first-order derivatives and positive-semidefinite Hessians (Wu et al., 2018):

$$\ell(\theta; x) = (\theta - \theta^*)^T \hat{H}(x)(\theta - \theta^*) + O(\|\theta\|^3). \tag{2}$$

The dynamics of $\theta$ thus obeys Eq. (1). This type of minimum is of great importance in deep learning because modern neural networks are often "overparametrized," and overparametrized networks under gradient flow are observed to reach these interpolation minima easily.

The dynamics we consider is more general than that around an interpolation minimum because the Hessians $\hat{H}$ in Eq. (1) are allowed to have eigenvalues of both positive and negative signs, whereas Eq. (2) only allows for positive semidefinite $\hat{H}$. Therefore, the general solution of Eq. (1) also helps us understand Eq. (2) once we restrict the study to PSD Hessians. The types of saddle points that

---

[1] Definitions of stability found in (Wu et al., 2018; Wu & Su, 2023; Ma & Ying, 2021; Mulayoff et al., 2021) are slightly different from our Def. 2. However, they also lead to Prop. 2.

[2] Note that $H(x_t)$ is independent for different $t$ because the sampling of $x_t$ is independent. Also, note that $H$ does not depend on $\theta$ by the linear-dynamics approximation.

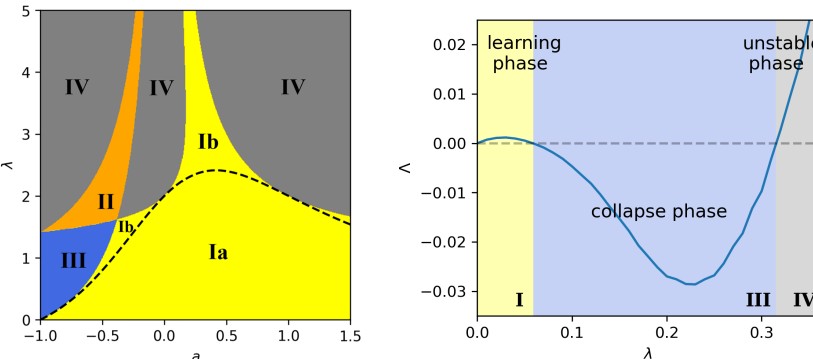

Figure 1: SGD exhibits a complex phase diagram through the lens of probabilistic stability. **Left**: $a$ denotes the parameter in the data distribution, as discussed in detail in section 4. For a matrix factorization saddle point, the dynamics of SGD can be categorized into at least five different phases. Phase **I**, **II**, and **IV** correspond to a successful escape from the saddle. Phase **III** is where the model converges to a low-rank saddle point. Phase **I** corresponds to the case $w_t \to_p u_t$, which signals correct learning. In phase **Ia**, the model also converges in variance. Phase **II** corresponds to stable but incorrect learning, where $w_t \to_p -u_t$. Phase **IV** corresponds to complete instability. **Right**: the phases of SGD can quantified by the sign of the Lyapunov exponent $\Lambda$. Where $\Lambda < 0$, SGD collapses to a saddle point; when $\Lambda > 0$, SGD escapes the saddle and enters a escaping phase. The two escaping phases are qualitatively different. For a small learning rate, the model is in a learning phase due to the repulsiveness of the saddle point at a small learning rate, and the model is likely to converge to local minima close to the saddle. For a very large learning rate, SGD escapes the saddle due to the dynamical instability of SGD, and the model will move far away from the saddle. Besides, the magnitude of the Lyapunov exponent can also quantity the speed of the learning dynamics. See Appendix A.1 for numerical details of this example.

obey Eq.(1) have been found to widely exist when there is any type of loss function symmetry, such as permutation symmetry (Fukumizu & Amari, 2000; Simsek et al., 2021; Entezari et al., 2021; Hou et al., 2019), rescaling symmetry (Dinh et al., 2017; Neyshabur et al., 2014), and rotation symmetry (Ziyin et al., 2023). Recently, it has been shown that every symmetry in the loss function leads to a critical point[3] of this type, and, more importantly, the subset of parameters relevant to the symmetry, up to the leading order, has a dynamics completely independent of the rest of the parameters (Ziyin, 2023). This means that Eq. (1) can be seen as an effective description for only a small subset of all parameters in the model under consideration and is not as restrictive as it naively seems to be.[4]

## 3 THE PROBABILISTIC STABILITY OF SGD

This section presents the main theoretical results. We first show that the dynamics is exactly solvable for a rank-1 dynamics. We then prove a result showing that no moment-based stability can understand the stability of SGD around the saddle. Lastly, we prove that the probabilistic stability of SGD is equivalent to a condition on the sign of the Lyapunov exponent of the dynamics.

### 3.1 RANK-1 DYNAMICS

Let us first consider the case in which $\hat{H}(x) = h(x)nn^T$ is rank-1 for a random scalar function $h(x)$, and a fixed unit vector $n$ for all data points $x$. Thus, the dynamics simplifies to a one-dimensional dynamics, where $h(x) \in \mathbb{R}$ is the corresponding eigenvalue of $\hat{H}(x)$:

$$\theta_{t+1} = \theta_t - \lambda h(x)(\theta_t - \theta^*). \tag{3}$$

**Theorem 1.** *Let $\theta_t$ follow Eq. (3). Then, for any distribution of $h(x)$, $n^T(\theta_t - \theta^*) \to_p 0$ if and only if*

$$\mathbb{E}_x[\log|1 - \lambda h(x)|] < 0. \tag{4}$$

The condition (4) is a sharp characterization of when a critical point becomes attractive. It also works with weight decay. When weight decay is present, the diagonal terms of $\hat{H}$ are shifted by $\gamma$, and so $h = h' + \gamma$.

---

[3]Note that this does not have to be a "point," but a critical submanifold. See an example of such a manifold due to permutation symmetry in Simsek et al. (2021).

[4]See proposition 3 for an example of how different directions have dynamics independent of the rest.

At what learning rate is the condition violated? To leading orders in $\lambda$, this can be identified by expanding the logarithmic term up to the second order in $\lambda$:

$$\mathbb{E}_x[\log|1 - \lambda h(x)|] = -\lambda\mathbb{E}[h(x)] - \frac{1}{2}\lambda^2\mathbb{E}_x[h(x)^2] + O(\lambda^3). \tag{5}$$

Ignoring the second-order term, we see that the dynamics always follow the sign of $\mathbb{E}[h(x)]$, in agreement with the GD algorithm. If $\mathbb{E}[h]<0$, the condition is always violated. When the second-order term is taken into consideration, the fluctuation of $h(x)$ now decides the stability of the stationary condition. The stationary condition is attractive if

$$\lambda > 2\frac{-\mathbb{E}[h(x)]}{\mathbb{E}[h(x)^2]}. \tag{6}$$

This result implies that the stationary condition can be attractive even if $h < 0$. The r.h.s. of the condition also has a natural interpretation as a signal-to-noise ratio in the gradient. The numerator is the Hessian of the original loss function, which determines the signal in the gradient. The denominator is the strength of the gradient noise in the minibatch (Wu et al., 2018). An illustration of this solution is given in Figure 1. We show the probabilistic stability conditions for a rank-1 saddle point with a rescaling symmetry (see Section 4). The loss function is $\ell(u, w) = -xyuw + o(u^2 + w^2)$. Here, the data points $xy = 1$, and $xy = a$ are sampled with equal probability. These saddles appear naturally in matrix factorization problems and also in recent sparse learning algorithms (Poon & Peyré, 2021; 2022; Ziyin & Wang, 2023; Kolb et al., 2023).

When the dynamics is high-dimensional, the problem becomes harder to solve because for each realization of $x$, $\hat{H}(x)$ do not commute with each other. While an analytical solution to the stability condition in high-dimension is unlikely to exist, we can say something quite general about them.

## 3.2 Insufficiency of Norm-Stability

Theorem 1 provides a perfect example to compare the probabilistic stability with the norm-stability. The following rather trivial proposition shows that if SGD converges to a point in $L_p$-norm, it must converge in probability.

**Proposition 1.** *If $\theta_t$ is stable at $\theta^*$ in $L_p$ norm, then it is stable at $\theta^*$ in probability.*

The proof follows from that convergence in $L_p$ norm implies the convergence in probability. Thus, norm stability is a more restricted notion than probabilistic stability. In many cases, the two types of stabilities agree. However, we will see that for SGD, the two types of stability conditions can offer dramatically different predictions, which is constructively established by the following proposition.

**Proposition 2.** *Let $\theta_t$ follow Eq. (1) around a critical point $\theta^*$. Then, for any fixed $\lambda$,*

1. *there exists a data distribution such that $\theta_t$ is probabilistically stable but not $L_p$-stable;*
2. *if $\theta^*$ is a saddle point and $p \geq 1$, the set of $\theta_0$ that is $L_p$-stable has Lebesgue measure zero.*

Therefore, this means that the $L_p$-stability is not useful in understanding the stability of SGD close to saddle points. One reason is that the outliers strongly influence the $L_p$ norm in the data, whereas the probabilistic stability is robust against such outliers. As discussed in detail in section 5 and shown in Fig. 6, the notion of probabilistic stability predicts the trajectory of learning qualitatively while $L_p$-stability fails in doing so.

## 3.3 Lyapunov Exponent and Probabilistic Stability

Extending the probabilistic stability to high-rank dynamics is nontrivial because the stability of SGD is generally initialization dependent, unlike the rank-1 case, where the condition is found to take an exact form and is initialization independent. It is thus useful to consider the worst-case initialization. Here, the crucial quantity is the *Lyapunov exponent* of a point $\theta^*$:[5]

$$\Lambda = \max_{\theta_0} \lim_{t \to \infty} \frac{1}{t}\mathbb{E}\left[\log\frac{\|\theta_t - \theta^*\|}{\|\theta_0\|}\right]. \tag{7}$$

---

[5]More appropriately, this should be called the maximum Lyapunov exponent, which is initialization-independent. One can also consider the initialization-dependent Lyapunov exponent. If $\bar{H}$ is a $d$-by-$d$ matrix, a well-known fact is that the initialization-dependent Lyapunov exponent takes at most $d$ distinctive values. Conceptually, this means that there can be, at most, $d$ collapses at different critical learning rates.

Here, the expectation is taken over the random samplings of the SGD algorithm. In general, $\Lambda$ does not vanish. The following theorem shows that SGD is probabilistically stable at a point if and only if its Lyapunov exponent is negative.

**Theorem 2.** *Assuming that $\Lambda \neq 0$, the linearized dynamics of SGD is probabilistically stable at $\theta^*$ for any $\theta_0$ if and only if $\Lambda < 0$.*

The proof of this theorem invokes the Furstenberg-Kesten theorem (Furstenberg & Kesten, 1960). It is straightforward to show that the Lyapunov exponent exists and to find an upper and lower bound. For a finite-size dataset, we can define $r_{\max}$ to be larger than the absolute value of the eigenvalues of $I - \lambda \hat{H}(x)$ for all $x$. Similarly, we can define $r_{\min} > 0$ to be smaller than all the absolute values of all the eigenvalues of $I - \lambda \hat{H}(x)$ for all $x$. Therefore, it is easy to check that

$$\log r_{\min} < \Lambda < \log r_{\max}. \tag{8}$$

However, it is difficult to give a better estimation of the exponent. In fact, it is a well-known open problem in the field of dynamical systems to find an analytical expression of the Lyapunov exponent (Crisanti et al., 2012; Pollicott, 2010; Jurga & Morris, 2019).

Now, we give two quantitative estimates about when the Lyapunov exponent will be negative. This discussion also implies a sufficient but weak condition for a general type of multidimensional dynamics to converge in probability. Let $h^*(x)$ be the largest eigenvalue of $\hat{H}(x)$ and assume that $1 - h^*(x) > 0$ for all $x$. Then, the following condition implies that $\theta \to_p 0$: $\mathbb{E}_x[\log|1 - \lambda h^*(x)|] < 0$, which mimics the condition we found for rank-1 systems. An alternative estimate can be made by assuming that $\hat{H}(x)$ commute with $\hat{H}(x')$ for all $x$ and $x'$. If $\hat{H}$ has rank $d$, this reduces the problem to $d$ separated rank-1 dynamics, and Theorem 1 gives the exact solutions in each subspace. Numerical evidence shows that the commutation approximation quite accurately predicts the onset of low-rank behavior the actual rank (see Section 4 and Appendix A.4).

Another relevant question is whether this theorem is trivial for SGD at a high dimension in the sense that it could be the case that $\Lambda$ could be identically zero independent of the dataset. One can show that the Lyapunov exponent is generally nonzero for all datasets that satisfy a mild condition. Let $\mathbb{E}[\hat{H}]$ be full rank. By definition,

$$\Lambda = \lim_{t \to \infty} \frac{1}{t} \mathbb{E}\left[ \log \theta_0^T \left( \prod_j^t (I - \lambda \hat{H}_{i_j}) \right) \left( \prod_j^t (I - \lambda \hat{H}_{i_j}) \right)^T \theta_0 \right] = -\frac{2\lambda \theta_0^T H \theta_0}{\|\theta_0\|^2} + O(\lambda^2). \tag{9}$$

Therefore, as long as $\lambda$ is sufficiently small, the sign of the Lyapunov exponent is opposite to the sign of the eigenvalues of $H$. This proves something quite general for SGD at an interpolation minimum: with a small learning rate, the model converges to the minimum exponentially fast, in agreement with common analysis in the optimization literature. See Figure 1-right for numerical computation of Lyapunov exponents of a matrix factorization problem and the corresponding phases.

## 4 PHASES OF LEARNING

With this notation of stability, we can study the actual effect of minibatch noise on a neural network-like landscape. The theory has interesting implications for the stability of interpolation minima in deep learning, which we explore in Appendix C.1. In the main text, we focus on the stability of SGD on saddle points.

A commonly studied minimal model of the landscape of neural networks is a deep linear net (or deep matrix factorization) (Kawaguchi, 2016; Lu & Kawaguchi, 2017; Ziyin et al., 2022a; Wang & Ziyin, 2022). For these problems, we understand that all local minima are identical copies of each other, and so all local minima have the same generalization capability (Kawaguchi, 2016; Ge et al., 2016). A deep linear net's special and interesting solutions are the saddle points, which are low-rank solutions and often achieve similar training loss with dramatically different generalization performances. More importantly, these saddles points also appear in nonlinear models with similar geometric properties, and they could be a rather general feature of the deep learning landscape (Brea et al., 2019). It is thus important to understand how the noise of SGD affects the stability of a low-rank saddle here. Let the loss function be $\mathbb{E}_x[(\sum_i u^{(i)}\sigma(w^{(i)}x) - y)^2/2]$, where $\sigma(x) = c_0 x + O(x^2)$ is any nonlinearity that is locally linear at $x = 0$. We let $c_0 = 1$ and focus on cases where both $x$ and

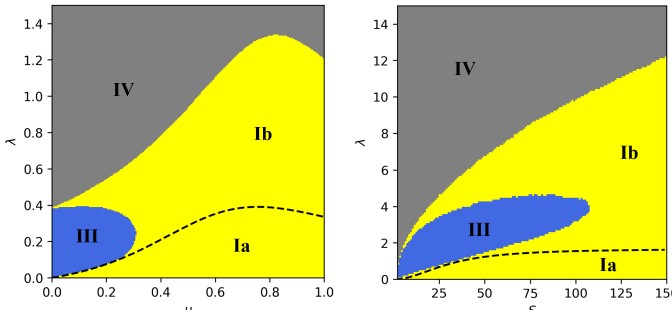

Figure 2: **Phase diagrams of SGD stability**. The definitions of the phases are the same as those in Figure 1. We sample a dataset of size $N$ such that $x \sim \mathcal{N}(0,1)$ and noise $\epsilon \sim \mathcal{N}(0,4)$, and generate a noisy label $y = \mu x + (1-\mu)\epsilon$. Left: the $\lambda - \mu$ (noise level) phase diagram for $S = 1$ and $N = \infty$. Right: The $\lambda - S$ (batch size) phase diagram for $\mu = 0.06$ and $N = \infty$.

$y$ are one-dimensional. Locally around $u$, $w \approx 0$, the model $u^T w$ is either rank-1 or rank-0. The rank-0 point where $u^{(i)} = w^{(i)} = 0$ for all $i$ is a saddle point as long as $\mathbb{E}[xy] \neq 0$. In this section, we show that the stability of this saddle point features complex and dramatic phase transition-like behaviors as we change the learning rate of SGD.

Consider the linearized dynamics around the saddle at $w^{(i)} = u^{(i)} = 0$. The expanded loss function takes the following form:

$$\ell(u, w; x, y) = -xy \sum_i^d u^{(i)} w^{(i)} + const. \tag{10}$$

For learning to happen, SGD needs to escape from the saddle point. Let us consider a simple data distribution where $xy = 1$ and $xy = a$ with equal probability. When $a > -1$, *correct* learning happens when $\text{sign}(w) = \text{sign}(u)$. We thus focus on the case of $a > -1$. The case of $a < -1$ is symmetric to this case up to a rescaling. This example is already presented in Figure 1. There are five phases of learning in this simple example

- Ia. correct learning with prob. and norm stability ($w_t - u_t \rightarrow_{L_2} 0$, $w_t + u_t$ diverges);
- Ib. correct learning with prob. but not norm stability ($w_t - u_t \rightarrow_p 0$, $w_t - u_t \not\rightarrow_{L_2} 0$, $w_t + u_t$ diverges);
- II. incorrect learning under probabilistic stability ($w_t - u_t$ diverges, $w_t + u_t \rightarrow_p 0$);
- III. convergence to low-rank saddle point ($w_t - u_t \rightarrow_p 0$, $w_t + u_t \rightarrow_p 0$);
- IV. completely unstable ($w_t + u_t$, $w_t - u_t$ diverges in p.).

The two most important observations are: (1) SGD can indeed converge to low-rank saddle points; however, this happens only when the gradient noise is sufficiently strong and when the learning rate is large (but not too large); (2) the region for convergence to saddles (region III) is exclusive with the region for convergence in mean square (Ia), and thus one can only understand the saddle-seeking behavior of SGD within the proposed probabilistic framework. Let $B$ denote a mini-batch and $S$ be its cardinality. We prove the following proposition.

**Proposition 3.** *For any $w_0, u_0 \in \mathbb{R}/\{0\}$. $w_t - u_t \rightarrow_p 0$ if and only if $\mathbb{E}_B\left[\log\left|1 - \lambda\sum_{(x,y)\in B} xy/S\right|\right] < 0$. $w_t + u_t$ converges to $0$ in probability if and only if $\mathbb{E}_B\left[\log\left|1 + \lambda\sum_{(x,y)\in B} xy/S\right|\right] < 0$.*

The theory shows that the phase diagram of SGD strongly depends on the data distribution, and it is interesting to explore and compare a few different settings. Now, we consider a size-$N$ Gaussian dataset. Let $x_i \sim \mathcal{N}(0,1)$ and noise $\epsilon_i \sim \mathcal{N}(0,4)$, and generate a noisy label $y_i = \mu x_i + (1-\mu)\epsilon_i$. See the phase diagram for this dataset in Figure 2 for an infinite $N$. The phase diagrams in Figure 7 show the phase diagram for a finite $N$. We see that the phase diagram has a very rich structure at a finite size. We make three rather surprising observations about the phase diagrams: (1) as $N \rightarrow \infty$, the phase diagram becomes smoother and smoother and each phase takes a connected region (cf. finite size experiments in Appendix A.2); (2) phase II seems to disappear as $N$ becomes large; (3) the lower part of the phase diagram seems universal, taking the same shape for all samplings of the datasets and across different sizes of the dataset. This suggests that the convergence to low-rank structures can be a universal aspect of SGD dynamics, which corroborates the widely observed phenomenon of collapse in deep learning (Papyan et al., 2020; Wang & Ziyin, 2022; Tian, 2022).

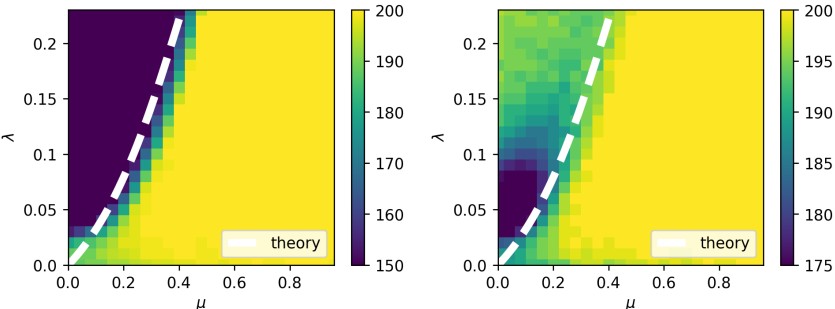

Figure 3: Convergence to low-rank solutions in nonlinear neural networks. At every training step, we sample input $x \sim \mathcal{N}(0, I_{200})$ and noise $\epsilon \sim \mathcal{N}(0, \sqrt{2}I_{200})$, and generate a noisy label $y = \mu x + (1 - \mu)\epsilon$, where $1 - \mu$ controls the level of the noise. We compute the rank of the second layer of the weight matrix after training. **Left**: Linear network. **Right**: tanh network. The white dashed line shows the theoretical prediction of the appearance of low-rank structure computed by numerically integrating the condition in Proposition 3.

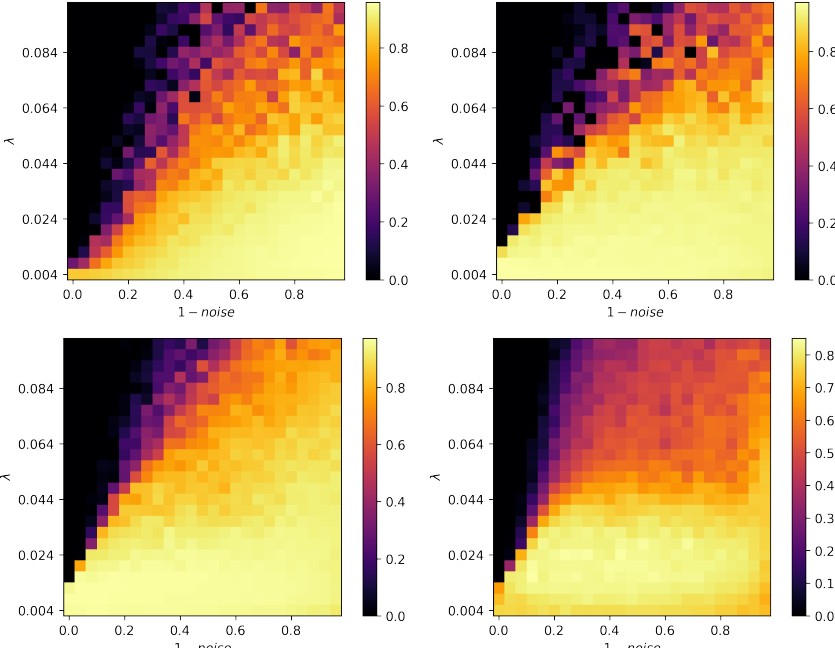

Figure 4: Density ($1-sparsity$) of the convolutional layers in a ResNet18, when there is static noise (mislabeling) in the training data. Here, we show the number of sparse parameters in the two of the largest convolutional layers, each containing roughly one million parameters in total. The figures respectively show layer1.1.conv2 (upper left), layer2.1.conv2 (upper right), layer3.1.conv2 (lower left), and layer4.1.conv2 (lower right).

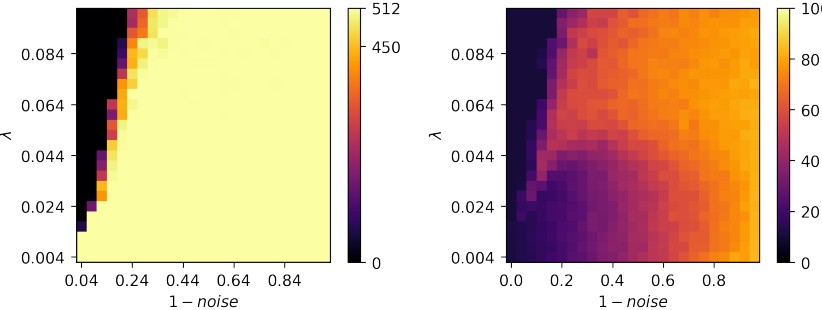

Figure 5: Rank (**left**) and test accuracy (**right**) of the ResNet18 trained in a data set with static noise. The transition of rank has a clear boundary. The model has a full rank but random-guess level performance for large noise and small learning rates. Here, *noise* refers to the probability that data point is mislabeled.

The theory also shows that if we fix the learning rate and noise level, increasing the batch size makes it more and more difficult to converge to the low-rank solution (see Figure 8, for example). This is expected because the larger the batch size, the smaller the effective noise in the gradient.

Many recent works have suggested how neural networks could be biased toward low-rank solutions. Theoretically, Galanti et al. (2023) showed that with weak weight decay, SGD is biased towards low-rank solutions. Ziyin et al. (2022a) showed that GD converges to a low-rank solution with weight decay. Therefore, weight decay already induces a low-rank bias in learning, and it is unknown if SGD alone has any bias toward low-rank solutions. Andriushchenko et al. (2022) showed empirical hints of a preference for low-rank solutions when training without SGD. However, it remains to be clarified when or why SGD has such a preference on its own. To the best of our knowledge, our theory is the first to precisely characterize the low-rank bias of SGD in a deep learning setting. Compared with the stability diagram of linear regression, this result implies that a large learning rate can both help and hinder optimization.

**Phase Diagram for Neural Networks**   There is a strong sense of universality in the lower-left part of the phase diagram in Figure 1 since they all take a similar shape independent of the size or sampling of the data points. We now verify its existence in actual neural networks.

We start with a controlled experiment where, at every training step, we sample input $x \sim \mathcal{N}(0, I_{200})$ and noise $\epsilon \sim \mathcal{N}(0, 4I_{200})$, and generate a noisy label $y = \mu x + (1 - \mu)\epsilon$. Note that $1 - \mu$ controls the level of the noise. Training proceeds with SGD on the MSE loss. We train a two-layer model with the architecture: $200 \to 200 \to 200$. See Figure 3 for the theoretical phase diagram, which is estimated under the commutation approximation. Under SGD, the model escapes from the saddle with a finite variance to the right of the dashed line and has an infinite variance to its left. In the region $\lambda \in (0, 0.2)$, this loss of the $L_2$ stability condition coincides with the condition for the convergence to the saddle. The experiment shows that the theoretical boundary agrees well with the numerical results.[6]

Lastly, we train independently initialized ResNets on CIFAR-10 with SGD. The training proceeds with SGD without momentum at a fixed learning rate and batch size $S = 32$ (unless specified otherwise) for $10^5$ iterations. Our implementation of Resnet18 contains 11M parameters and achieves $94\%$ test accuracy under the standard training protocol, consistent with the established values. To probe the effect of noise, we artificially inject a dynamical label noise during every training step, where, at every step, a correct label is flipped to a random label with probability $noise$, and we note that the phase diagrams are similar regardless of whether the noise is dynamical or static (where the mislabelling is fixed). See Figure 5 for the phase diagram of static label noise. Interestingly, the best generalization performance is achieved close to the phase boundary when the noise is strong. This is direct evidence that SGD noise has a strong regularization effect on the trained model. We see that the results agree with the theoretical expectation and the analytical model's phase diagram. We also study the sparsity of the ResNets in different layers in Figure 4, and we observe that the phase diagrams are all qualitatively similar. Also, see Appendix A for the experiment with a varying batch size.

## 5   WHICH SOLUTION DOES SGD PREFER?

We now investigate one of the most fundamental problems in deep learning through the lens of probabilistic stability: how SGD selects a solution for a neural network. In this section, we study a two-layer network with a single hidden neuron with the swish activation function: $f(w, u, x) = u \times \text{swish}(wx)$, where $\text{swish}(x) = x \times \text{sigmoid}(x)$. Swish is a differentiable ReLU variant discovered by meta-learning techniques and consistently outperforms ReLU in various tasks. We generate 100 data points $(x, y)$ as $y = 0.1\text{swish}(x) + 0.9\epsilon$, where both $x$ and $\epsilon$ are sampled from normal distributions. See Figure 6 for an illustration of the training loss landscape. There are two local minima: solution A at roughly $(-0.7, -0.2)$ and solution B at $(1.1, -0.3)$. Here, the solution with better generalization is A because it captures the correct correlation between $x$ and $y$ when $x$ is small. Solution A is also the sharper one; its largest Hessian eigenvalue is roughly $h_a = 7.7$. Solution B is the worse solution; it is also the flatter one, with the largest Hessian value being $h_b = 3.0$. There is

---

[6]The Adam optimizer (Kingma & Ba, 2014) also have a similar phase diagram. See Appendix A. This suggests that the effects we studied are rather universal, not just a special feature of SGD.

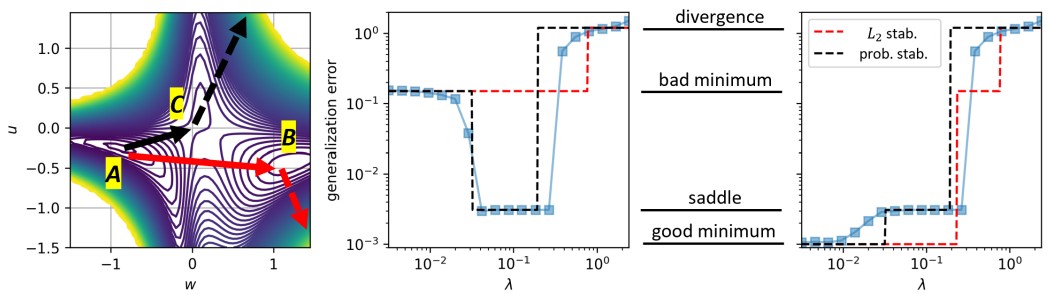

Figure 6: **How SGD selects a solution**. **Left**: The landscape of a two-layer network with the swish activation function (Ramachandran et al., 2017). The black arrow corresponds to the experimental trajectory and the prediction of probabilistic stability, while the red arrow corresponds to the (false) prediction of the $L_2$ stability. **Middle, Right**: the generalization performance of the model for different learning rates. **Middle**: Initialized at solution B, SGD first jumps to C and then diverges. **Right**: Initialized at A, SGD also jumps to C and diverges. In both cases, the behavior of SGD agrees with the prediction of the probabilistic stability instead of the $L_2$ stability. Instead of jumping between local minima, SGD, at a large learning rate, transitions from minima to saddles.

also a saddle point C at $(0, 0)$, which performs significantly better than B and slightly worse than A in generalization.

If we initialize the model at A, $L_2$ stability theory would predict that as we increase the learning rate, the model moves from the sharper solution A to the flatter minimum B when SGD loses $L_2$ stability in A; the model would then lose total stability once SGD becomes $L_2$-unstable at B. As shown by the red arrows in Figure 6. In contrast, probabilistic stability predicts that SGD will move from A to C as C becomes attractive and then lose stability, as the black arrows indicate. See the right panel of the figure for the comparison with the experiment for the model's generalization performance. The dashed lines show the predictions of the $L_2$ stability and probabilistic theories, respectively. We see that the probabilistic theory predicts both the error and the place of transition right, whereas $L_2$ stability neither predicts the right transition nor the correct level of performance.

If we initialize at B, the flatter minimum, $L_2$ stability theory would predict that the model will only have one jump from B to divergence as we increase the learning rate. Thus, from $L_2$ stability, SGD would have roughly the performance of B until it diverges, and having a large learning rate will not help increase the performance. In sharp contrast, the probabilistic stability predicts that the model will have two jumps: it stays at B for a small $\lambda$ and jumps to C as it becomes attractive at an intermediate learning rate. The model will ultimately diverge if C loses stability. Thus, our theory predicts that the model will first have a bad performance, then show a better performance at an intermediate learning rate, and finally diverge. See the middle panel of Figure 6. We see that the prediction of the probabilistic stability agrees with the experiment and correctly explains why SGD leads to better performance.

## 6 DISCUSSION

In this work, we have demonstrated that the convergence in probability condition serves as an essential notion for understanding the stability of SGD close to saddle points. Crucially, these effects are only present for SGD and not for GD, demonstrating that the algorithmic regularization due to SGD is qualitatively different from that of GD. We also clarified its intimate connection to Lyapunov exponents, which are fundamental metrics of stability in the study of dynamical systems. The proposed stability agrees with the norm-based notion of stability at a small learning rate and large batch size. At a large learning rate and a small batch size, we have shown that the proposed notion of stability captures the actual behavior of SGD much better and successfully explains a series of experiment phenomena that have been quite puzzling. Among the many implications that we discussed, perhaps the most fundamental one is a novel understanding of the implicit bias of SGD. When viewed from a dynamical stability point of view, the implicit bias of stochastic gradient descent is thus fundamentally different from the implicit bias of gradient descent. In the proposed perspective, SGD performs a selection between converging to saddles and to local minima, not between sharp minima and flat ones.

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

# A  ADDITIONAL NUMERICAL RESULTS

## A.1  EXPERIMENTAL DETAIL FOR FIGURE 1-RIGHT

The experiment is performed for a two-dimensional system whose dynamics is specified in (1). The expectation of the Hessian $\mathbb{E}[\hat{H}]$ is chosen to be $\text{diag}(0.1, -0.1)$, while the noise is generated via a normal $2 \times 2$ random matrix $M_{\text{noise}}$. The noisy Hessian is obtained as

$$\hat{H} = \mathbb{E}[\hat{H}] + M_{\text{noise}} + M_{\text{noise}}^T, \tag{11}$$

and one can verify that such $\hat{H}$ is symmetric and consistent with our choice of $\mathbb{E}[\hat{H}]$. The initial state is sampled from a unit circle. The dynamics stops at time step $t$, and the Lyapunov exponent is calculated as $\frac{1}{t} \log \|\theta_t\|$, if one of the three following conditions is satisfied: $\|\theta\|$ reaches the upper cutoff of $10^{100}$; $\|\theta\|$ reaches the lower cutoff of $10^{-140}$; the preset maximal number of steps of $5000$ is reached. For each learning rate, the Lyapunov is obtained as the average of the results collected in $800$ independent runs.

## A.2  PHASES OF FINITE-SIZE DATASETS

See Figure 7.

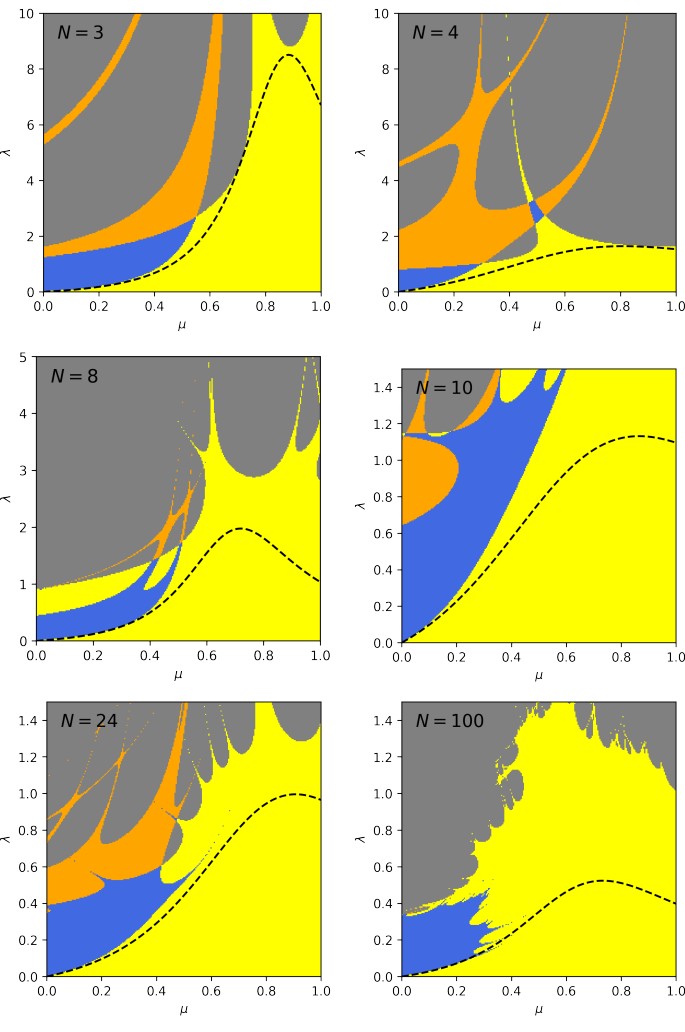

Figure 7: **Phase diagrams of SGD stability for finite-size dataset**. The data sampling is the same as in Figure 2. From upper left to lower right: $N = 3, 4, 8, 10, 24, 100$. As the dataset size tends to infinity, the phase diagram converges to that in Figure 2. The lower parts of all the phase diagrams look similar, suggesting a universal structure.

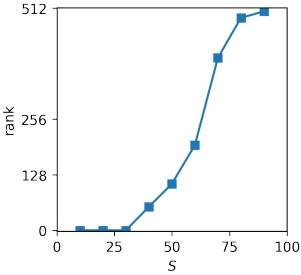

Figure 8: The rank of the penultimate-layer representation of Resnet18 trained with different levels of batch sizes. In agreement with the phase diagram, the model escapes from the low-rank saddle as one increases the batch size.

## A.3 EFFECT OF CHANGING BATCHSIZE ON RESNET18

See Figure 8.

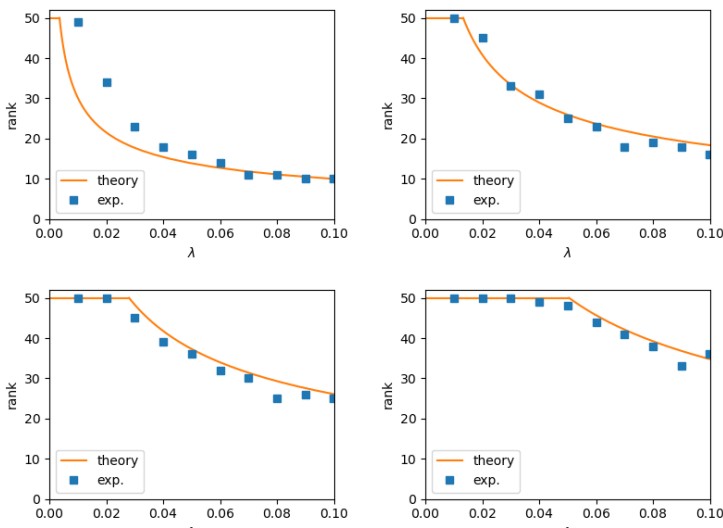

Figure 9: Rank vs learning rate in a vanilla matrix factorization problem for, from upper left to lower right, $\mu = 0.05,\ 0.15,\ 0.25,\ 0.35$. The theoretical curve is from the commutation approximation where each subspace of the model collapses at the critical learning rate $\lambda = -2\frac{\mathbb{E}[h(x)]}{\mathbb{E}[h^2(x)]}$.

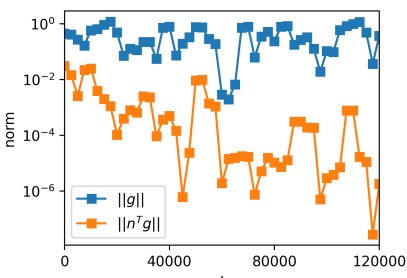

Figure 10: Norms of gradient ($g$) of a matrix factorization problem trained with SGD. $n$ is a low-rank direction after training. During training, it is commonly the case the model does not converge to a stationary point but to a stationary distribution. Our theory is compatible with this case because it is possible and common for the model to converge to a point in some subspace, even if it is not converging to a point overall.

## A.4 Commutation Approximation

Here, we compare the empirical rank of the solution with the commutation approximated critical learning rates obtained in (6). See Figure 9. The experiment is run on a two-layer fully connected linear network: $50 \to 50 \to 50$, which is equivalent to a matrix factorization problem. The model is initialized with the standard Kaiming init. The dataset we consider is one with a sparse but full-rank signal.

Let $\odot$ denote the Hadamard product. The input data is generated as $x = m \odot X$, where $X \sim \mathcal{N}(0, I_{50})$ and $m$ is a random mask where a random element is set to be 1, and the rest is zero. The labels $Y$ is generated as $Y = \mu x + (1 - \mu)(m \odot \epsilon)$, where $\epsilon \sim \mathcal{N}(0, 2\mathrm{diag}(0.01, 0.05, ...., 2.01))$ is the noise.

## A.5 Convergence to Stationary distributions

As we mentioned in the main text, our theory is compatible with the case when the model converges to a stationary distribution but not a stationary point, which is more commonly the case during actual

deep learning practice (Zhang et al., 2022). The experimental setup is the same as in the previous section.

See Figure 10, where we plot the norm of the gradient $g$ and the norm of $n^T g$, where $n$ is a low-rank direction after training. Here, we see that the norm of the gradient does not converge to zero, but to a positive value, signaling a convergence to a stationary distribution. At the same time, the norm of $n^T g$ does converge to zero, which means that in some subspace, the parameters do converge to a point. This justifies our argument.

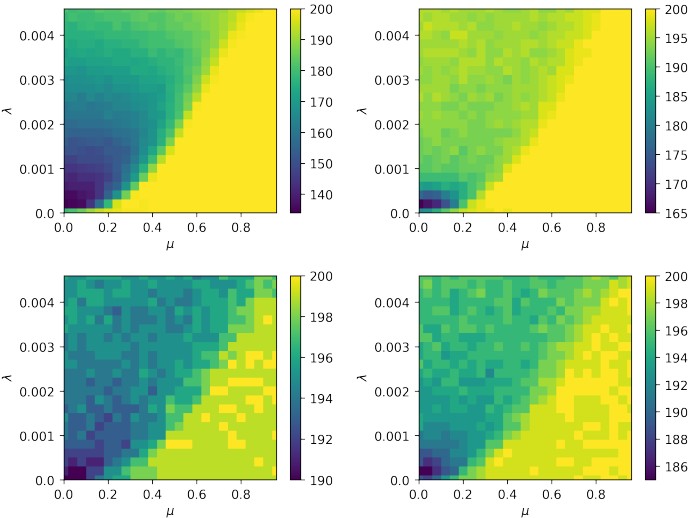

Figure 11: Rank of the converged solution for two-layer linear (upper left), tanh (upper right), relu (lower left) and swish (lower right) models.

# B    EXPERIMENT WITH ADAM

## B.1    EXPERIMENT WITH ADAM

We note that the phenomena we studied is not just a special feature of the SGD, but, empirically, seems to be a universal feature of first-order optimization methods that rely on minibatch sampling. Here, we repeat the experiment in Figure 2. We train with the same data and training procedure, except that we replace SGD with Adam (Kingma & Ba, 2014), the most popular first-order optimization method in deep learning. Figure 11 shows that similarly to SGD, Adam also converges to the low-rank saddles in similar regions of learning rate and $\mu$.

# C  ADDITIONAL THEORETICAL CONCERNS

## C.1  A SIMPLE EXAMPLE OF INTERPOLATION MINIMUM

To illustrate different concepts of stability of the SGD algorithm, we examine a simple one-dimensional linear regression problem. The training loss function for this problem is defined as $L(w) = \frac{1}{N} \sum_i^N (wx_i - y_i)^2$.

For GD, the dynamics diverges when the learning rate is larger than twice the inverse of the largest Hessian eigenvalue. To see this, let $H = \mathbb{E}_x[\hat{H}(w^*, x)]$ denote the Hessian of $L$ and $h$ its largest eigenvalue. Using GD leads to $\|w_{t+1}\| = \|w_0(I - \lambda H)^t\| \propto |1 - \lambda h|^t$. Divergence happens when $|1 - \lambda h| > 1$. The range of viable learning rates is thus:

$$\lambda_{\text{GD}} \leq 2/h = 2/\mathbb{E}_x[x^2]. \tag{12}$$

Naively, one would expect that a similar condition approximates the stability condition for the case when mini-batch sampling is used to estimate the gradient.

For SGD, the variance stability condition is the same as the condition that the second moment of SGD decreases after every time step, starting from an arbitrary initialization (see Section C.1.1):

$$\lambda_{\text{DS}} \leq \frac{2S^2 \mathbb{E}[x^2]}{\mathbb{E}[x^4] + (S-1)^2 \mathbb{E}[x^2]^2}. \tag{13}$$

Also related is the stability condition proposed by Ziyin et al. (2022b), who showed that starting from a stationary distribution, $w$ stays stationary under the condition $\lambda_{\text{SS}} < \frac{2}{h} \frac{1}{1+1/S}$, which we call the stationary linear stability condition (SS). When we have batch size 1, the stability condition halves: $\lambda < 1/h$. For all stability conditions, we denote the maximum stable learning rate with an asterisk as $\lambda^*$.

For the probabilistic stability, let us consider the interpolation regime, where all data points $(x, y) \in \mathbb{R}^2$ lie on a straight line. In this situation, the loss function has a unique global minimum of $w^* = y_i/x_i$ for any $i$. Applying Theorem 1, one can immediately prove the following proposition.

**Proposition 4.** *Let $\lambda$ be such that $\mathbb{E}_x[\log|1 - \lambda x^2|] \neq 0$. Then, for any $w_0$, $w_t \to_p w^*$ if and only if $\mathbb{E}_x[\log|1 - \lambda x^2|] < 0$.*

It is worth remarking that this condition is distinctively different from the case in which the gradient noise is a parameter-independent random vector. For example, Liu et al. (2021) showed that if the gradient noise is a parameter-independent Gaussian, SGD diverges in distribution if $\lambda > 2/h$. This suggests that the fact that the noise of SGD is $w$-dependent is crucial for its probabilistic stability.

One of the implications of the probabilistic stability is that for $\lambda = 1/x_i^2$, the SGD dynamics is always stable. Therefore, the largest stable learning rate is roughly given by:

$$\lambda_{\text{max}} = 1/x_{\text{min}}^2. \tag{14}$$

However, for these special choices of learning rates, the moment stability is not always guaranteed. As mentioned earlier, convergence in mean occurs when $\lambda \leq \lambda_{DS}^*$. However, this condition does not hold when $\lambda = 1/x_{\text{min}}^2$ and $x_{\text{min}} < \mathbb{E}[x_i]/2$, which is often the case for standard datasets. This result shows that the maximal learning rate that ensures stable training can be much larger than the maximal learning rate required for convergence in mean (cf. (13)). For a fixed value of $\mathbb{E}[x^2]$, $x_{\text{min}}^2$ can be arbitrarily small, which means that the maximal stable learning rate can be arbitrarily large. Another consequence of this result is that the stability of SGD depends strongly on individual data points and not just on summary statistics of the whole dataset.

This result highlights the importance of the Lyapunov exponent $\Lambda = \mathbb{E}_x[\log|1 - \lambda x^2|]$ and its sign in understanding the convergence of $w_t$ to the global minimum. When $\Lambda$ is negative, the convergence to the global minimum occurs. If $\Lambda$ is positive, SGD becomes unstable. We can determine when $\Lambda$ is negative for a training set of finite size by examining the following equation:

$$\Lambda = \frac{1}{N} \sum_i \log|1 - \lambda x_i^2|, \tag{15}$$

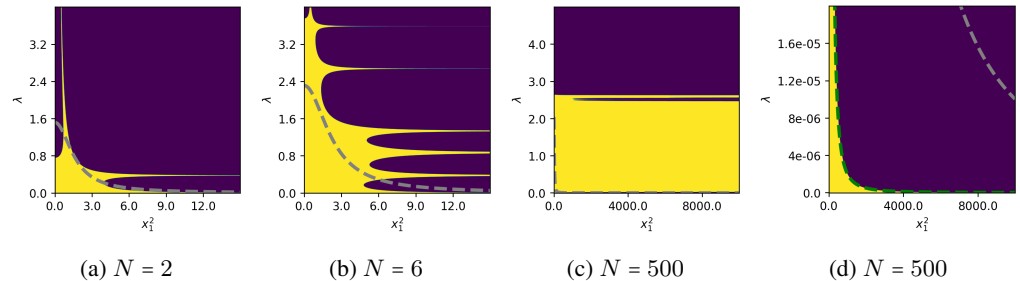

| (a) $N = 2$ | (b) $N = 6$ | (c) $N = 500$ | (d) $N = 500$ |

Figure 12: **Stability of SGD against a single outlier data** in a dataset of size $N$. Yellow denotes where SGD converges in probability, and dark blue denotes divergence. We control the norm of the first data point ($x_1^2$) while sampling the rest data from a standard normal distribution. (a-c) Stability of SGD for different sizes of the dataset; (d) zoom-in of (c) at a small learning rate. The grey dashed curves show $\lambda_{GD}^*$, and the green dashed curve shows $\lambda_{GD}^*/N$. The intermediate finite learning rates are robust against outliers in the data, whereas the smallest learning rates are strongly sensitive to outliers in the data.

which is negative when $\lambda$ is close to $1/x_i^2$ for some $i \in 1, \dots, N$. What is the range of $\lambda$ values that satisfy this condition? Suppose that $\lambda$ is in the vicinity of some $1/x_i^2$: $\lambda = \delta\lambda + 1/x_i^2$, and the instability is caused by a single outlier data point $x_{\text{out}} \gg 1$. Then, $\Lambda$ is determined by the competing contributions from the outlier, which destabilizes training, and $x_i^2$, which stabilizes training, and the resulting condition is approximately $|1 - \lambda x_i^2| < 1/|\lambda x_{\text{out}}^2|$. Because $\lambda \approx 1/x_i^2$, this condition leads to:

$$|\delta\lambda| < x_i^2/x_{\text{out}}^2. \tag{16}$$

This is a small quantity. However, if we change the learning rate to the stability region associated with another data point $x_j$ as soon as we exit the stability region of $x_i$, we still maintain stability. Therefore, the global stability region depends on the density of data points near $x_i$. Assuming that there are $N$ data points near $x_i$ with a variance of $\sigma^2$, the average distance between $x_i$ and its neighbors is approximately $\sigma^2/N$. As long as $\sigma^2/N < x_i^2/x_{\text{out}}^2$, SGD will remain stable in a large neighborhood. In practical terms, this means that when the number of data points is large, SGD is highly resilient to outliers in the data as shown in Figure 12. We see that the region of convergence in probability is dramatic, featuring stripes of convergent regions that correspond to $1/x_i^2$ for each data point and divergent regions where $\Lambda > 0$. While simple, this example has a fundamental implication: there are problems that cannot be learned by SGD at a small learning rate but can be learned by SGD at a finite learning rate. This implies the insufficiency of the commonly used stochastic differential equation theories of SGD (Li et al., 2021).

An important implication is the robustness of SGD to outliers in comparison to gradient descent. As Figure 12 shows, the bulk region of probabilistic stability stays roughly unchanged as the outlier data point becomes larger and larger; in contrast, both $\lambda_{GD}^*$ and $\lambda_{DS}^*$ decreases quickly to zero. In the bulk region of the learning rates, SGD is thus probabilistically stable but not stable in the moments. Meanwhile, in sharp contrast to this bulk robustness is the sensitivity of the smallest branch of learning rates of SGD to the outliers. Assuming that there is an outlier data point with a very large norm $c \gg N$, the largest $\lambda_{\text{GD}}$ scales as $\lambda_{\text{max}} \sim Nc^{-1}$. In contrast, for SGD, the smallest branch of the probabilistically stable learning rate scales as $c^{-1}$, independent of the dataset size. This means that if we only consider the smallest learning rate, SGD is much less stable than GD, and one needs to use a much smaller learning rate to ensure convergence. For $\lambda_{\text{DS}}$ a detailed analysis in Section C.1.1 shows that $\lambda_{\text{DS}}^* = (Nc)^{-1}$. Thus, the threshold of convergence in mean square is yet one order of magnitude smaller than that of probabilistic convergence. In the limit $N \to \infty$, SGD cannot converge in variance but can still converge in probability.

### C.1.1 $L_2$ STABILITY CONDITIONS AND THE DERIVATION OF EQ. (13)

For a general batch size $S$, the dynamics of SGD reads

$$w_{t+1} = w_t - \lambda w_t \frac{1}{S} \sum_{i=1}^{S} x_i^2 \tag{17}$$

$$= w_t \left( 1 - \lambda \frac{1}{S} \sum_{i=1}^{S} x_i^2 \right). \tag{18}$$

The second moment of $w_{t+1}$ is

$$\mathbb{E}_x[w_{t+1}^2 | w_t] = w_t^2 \mathbb{E}_x \left( 1 - \lambda \frac{1}{S} \sum_{i}^{S} x_i^2 \right)^2 \tag{19}$$

$$= w_t^2 \left( 1 - 2\lambda \mathbb{E}[x^2] + \frac{\lambda^2}{S^2} \sum_{i,j}^{S} \mathbb{E}[x_i^2 x_j^2] \right) \tag{20}$$

$$= w_t^2 \left( 1 - 2\lambda \mathbb{E}[x^2] + \frac{\lambda^2}{S^2} \mathbb{E}[x^4] + \frac{\lambda^2 (S-1)^2}{S^2} \mathbb{E}[x^2]^2 \right). \tag{21}$$

Note that this equation applies to any $w_t \in \mathbb{R}$. Therefore, the second moment of $w_t$ is convergent if

$$1 - 2\lambda \mathbb{E}[x^2] + \frac{\lambda^2}{S^2} \mathbb{E}[x^4] + \frac{\lambda^2 (S-1)^2}{S^2} \mathbb{E}[x^2]^2 < 1, \tag{22}$$

which solves to

$$\lambda < \frac{2S^2 \mathbb{E}[x^2]}{\mathbb{E}[x^4] + (S-1)^2 \mathbb{E}[x^2]^2}. \tag{23}$$

This condition applies to any data distribution. One immediate observation is that it only depends on the second and fourth moments of the data distribution and that both moments need to be finite for convergence at a non-zero learning rate. It is quite instructive to solve this condition under a few special conditions.

**Gaussian Data Distribution**  The condition (23) takes a precise form when the data is Gaussian. Using the fact that for a Gaussian variable $x$ with variance $\sigma^2$, $\mathbb{E}[x^4] = 3\sigma^4$, the condition simplifies to

$$\lambda < \frac{2}{\mathbb{E}[x^2]} \frac{S^2}{3S + (S-1)^2}. \tag{24}$$

This is the same as Eq. (13).

**Bernoulli Dataset**  Another instructive case to consider is the case when there are only two data points in the data: $x_1$ and $x_2$. The moments are

$$\begin{cases} \mathbb{E}[x^2] = \frac{1}{2}(x_1^2 + x_2^2), \\ \mathbb{E}[x^4] = \frac{1}{2}(x_1^4 + x_2^4). \end{cases} \tag{25}$$

When one of the data points, say $x_1$, is large, the condition becomes

$$\lambda < \frac{2S^2 x_1^2}{2x_1^4 + (S-1)^2 x_1^4} = \frac{2S^2}{x_1^2} \frac{1}{2 + (S-1)^2}. \tag{26}$$

**Extreme Outlier**  We can also consider the general case of a finite dataset with a large outlier, $x_{\max}$. The condition is similar to the Bernoulli case. We have

$$\begin{cases} \mathbb{E}[x^2] \approx \frac{1}{N} x_{\max}^2, \\ \mathbb{E}[x^4] \approx \frac{1}{N} x_{\max}^4. \end{cases} \tag{27}$$

The condition reduces to

$$\lambda < \frac{2S^2}{N x_{\max}^2} \frac{1}{1 + \frac{(S-1)^2}{N}}. \tag{28}$$

This can be seen as the generalization of the Bernoulli condition. When $S = 1$, this condition becomes

$$\lambda < \frac{2}{Nx_{\max}^2}. \tag{29}$$

There are many other interesting limits of this condition we can consider from the perspective of extreme value theory. However, this is beyond the scope of this work and we leave it as an interesting future work.

### C.2  PROOFS

#### C.2.1  PROOF OF THEOREM 1

*Proof.* Consider Eq. (1):

$$\theta_{t+1} = \theta_t - \lambda \hat{H}(x)(\theta_t - \theta^*). \tag{30}$$

Defining $w_t = \theta_t - \theta^*$, this equation can be written as

$$w_{t+1} = w_t - \lambda \hat{H}(x)w_t, \tag{31}$$

which is mathematically equivalent to the case when $\theta^* = 0$. Therefore, without loss of generality, we write the dynamics in the form of Eq. (31) in this proof and the rest of the proofs.

Now, when $\hat{H} \propto nn^T$ is rank-1, we can multiply $n^T$ from the left on both sides of the dynamics to obtain

$$n^T w_{t+1} = n^T w_t - \lambda h(x)n^T w_t. \tag{32}$$

The dynamics thus becomes one-dimensional in the direction of $n^T$.

Let $h_t$ denote the eigenvalue of the Hessian of the randomly sampled batch at time step $t$. The dynamics in Eq. (3) implies the following dynamics

$$\|n^T w_{t+1}\|/\|n^T w_t\| = |1 - \lambda h_t|, \tag{33}$$

which implies

$$\|n^T w_{t+1}\|/\|n^T w_0\| = \prod_{\tau=1}^{t} |1 - \lambda h_\tau|. \tag{34}$$

We can define auxiliary variables $z_t := \log(\|n^T w_{t+1}\|/\|n^T w_0\|) - m$ and $m := \mathbb{E}[\log(\|n^T w_{t+1}\|/\|n^T w_0\|)] = t\mathbb{E}_x[\log|1 - \lambda h_t|]$. Let $\epsilon > 0$. We have that

$$\mathbb{P}(\|n^T w_t\| < \epsilon) = \mathbb{P}(\|n^T w_0\|e^{z_t+m} < \epsilon) \tag{35}$$

$$= \mathbb{P}\left(\frac{1}{t}z_t < \frac{1}{t}(\log \epsilon/\|n^T w_0\| - m)\right) \tag{36}$$

$$= \mathbb{P}\left(\frac{z_t}{t} < -\mathbb{E}_x \log|1 - \lambda h_t| + o(1)\right). \tag{37}$$

By the law of large numbers, the left-hand side of the inequality converges to 0, whereas the right-hand side converges to a constant. Thus, we have, for all $\epsilon > 0$,

$$\lim_{t\to\infty} \mathbb{P}(\|n^T w_t\| < \epsilon) = \begin{cases} 1 & \text{if } m < 0; \\ 0 & \text{if } m > 1. \end{cases} \tag{38}$$

This completes the proof. $\qquad\square$

#### C.2.2  PROOF OF PROPOSITION 2

*Proof.* Part 2 of the proposition follows immediately from Proposition 5, which we prove below. Here, we prove part 1.

It suffices to consider a dataset with two data points for which $h(x_1) = 1/\lambda$ and $h(x_2) = c_0$, where each data point is sampled with equal probability. Let $c_0$ be such that

$$|1 - \lambda c_0|^p > \frac{1}{2}. \tag{39}$$

Now, we claim that this dynamics converges to zero in probability. To see this, note that

$$\|z_{t+1}\| = \begin{cases} \|z_t\| |1 - \lambda/\lambda| = 0 & \text{with probability } 0.5; \\ \|z_t\| |1 - \lambda c_0| & \text{with probability } 0.5. \end{cases} \tag{40}$$

Therefore, at time step $t$, $\mathbb{P}(z_t = 0) \geq 1 - 2^{-t}$, which converges to 0. This means that $z_t$ converges in probability to 0.

Meanwhile, the $p$-norm is

$$\mathbb{E}[\|z_{t+1}\|^p] = \frac{1}{2} \mathbb{E}[\|z_t\|^p] |1 - \lambda c_0|^p \tag{41}$$

$$\propto \frac{1}{2^t} |1 - \lambda c_0|^{pt} \to 0. \tag{42}$$

The convergence to zero follows from the construction that $|1 - \lambda c_0|^p > \frac{1}{2}$. This completes the proof. $\qquad\square$

**Proposition 5.** *(No convergence in $L_p$.) For every strict saddle point $\theta^*$, there exists an initialization $\theta_0$ such that for any $\lambda \in \mathbb{R}_+$ and distance function $f(\cdot, \theta^*)$, $\theta^*$ is unstable in $f$.*

*Proof.* This problem is easy to prove when $\theta$ is one-dimensional. For a high-dimensional $\theta$, the dynamics of SGD is

$$\theta_{t+1} = (I - \lambda \hat{H}_t)\theta_t. \tag{43}$$

Note that the expected value of $\theta_t$ is the same as the gradient descent iterations:

$$\mathbb{E}[\theta_{t+1}] = (I - \lambda \mathbb{E}[\hat{H}])\mathbb{E}[\theta_t] = (I - \lambda \mathbb{E}[\hat{H}])^t \theta_0, \tag{44}$$

which diverges if $\theta_0$ is in one of the escape directions of $\mathbb{E}[\hat{H}]$, which exist by the definition of strict saddle points.

Taking the $f$-distance of both sides and taking expectation, we obtain

$$\mathbb{E}[f(\theta_t, \theta^*)] \geq f(\mathbb{E}[\theta_t], \theta^*) \tag{45}$$

$$= f\left((I - \lambda \mathbb{E}[\hat{H}])^t \theta_0, \theta^*\right) \nrightarrow 0. \tag{46}$$

The first line follows from the fact that the distance function is convex by definition, and so one can apply Jensen's inequality.

Therefore, as long as $\theta_0$ overlaps with the concave directions of $\mathbb{E}[\hat{H}]$, the argument of $f$ diverges, which implies that the distance function converges to a nonzero value. The expected value of $\theta_t$ is just the gradient descent trajectory, which diverges for any strict saddle point.

By definition, $\mathbb{E}[\hat{H}]$ contains at least one negative eigenvalue, and so the directions that do not overlap with this direction are strict linear subspaces with dimension lower than the the total available dimensions. This is a space with Lesbegue measure zero. The proof is complete. $\qquad\square$

### C.2.3 Proof of Theorem 2

Let us first state the Furstenberg-Kesten theorem.

**Theorem 3.** *(Furstenberg-Kesten theorem) Let $X_1$, $X_2$, $X_3$, ... be independent random square matrices drawn from a metrically transitive time-independent stochastic process and $\mathbb{E}[\log_+ \|X^1\| < \infty]$, then[7]*

$$\lim_{n \to \infty} \frac{1}{n} \log \|X_1 X_2 ... X_n\| = \lim_{n \to \infty} \mathbb{E}\left[\frac{1}{n} \log \|X_1 X_2 ... X_n\|\right] \tag{47}$$

*with probability 1, where $\|\cdot\|$ denotes any matrix norm.*

Namely, the Lyapunov exponent of every trajectory converges to the expected value almost surely. Essentially, this is a law of large numbers for the Lyapunov exponent.

Now, we present the proof of Theorem 2.

---

[7]$\log_+(x) = \max(\log x, 0)$.

*Proof.* First of all, we define $m_t = \log \|\theta_t - \theta^*\|$ and $z_t = m_t - \mathbb{E}[m_t]$. By definition, we have

$$\mathbb{P}(g_t < \epsilon) = \mathbb{P}(e^{z_t + m_t} < \epsilon) \tag{48}$$

$$= \mathbb{P}\left(\frac{1}{t}(z_t + \mathbb{E}[m_t]) < \frac{1}{t}\log \epsilon\right) \tag{49}$$

$$= \mathbb{P}\left(\frac{1}{t}(z_t + \mathbb{E}[m_t]) < o(1)\right). \tag{50}$$

We can lower bound this probability by

$$\mathbb{P}\left(\frac{1}{t}(z_t + \mathbb{E}[m_t]) < o(1)\right) \geq \mathbb{P}\left(\frac{1}{t}\max_{\theta_0}(z_t + \mathbb{E}[m_t]) < o(1)\right). \tag{51}$$

By the definition of SGD, we have

$$\frac{1}{t}\max_{\theta_0}(z_t(\theta_0) + \mathbb{E}[m_t]) = \frac{1}{t}\max_{\theta_0}\log\left\|\prod_i^t (I - \lambda \hat{H}_i)(\theta_t - \theta_0)\right\|. \tag{52}$$

By the Furstenberg-Kesten theorem (Furstenberg & Kesten, 1960), this quantity converges to the constant $\Lambda = \lim_{t\to\infty} \mathbb{E}[m_t]/t \in \mathbb{R}$ almost surely. Namely, $z_t/t$ converges to 0 for almost every SGD trajectory.

Thus, for every $\epsilon$, if $\Lambda < 0$, Eq. (50) can be bounded as

$$\lim_{t\to\infty} \mathbb{P}(g_t < \epsilon) = \mathbb{P}(\Lambda < 0) = 1. \tag{53}$$

Because $\Lambda$ is a constant, we have that if $\Lambda < 0$, all trajectories from all initialization converge to 0. This finishes the first part of the proof. For the second part, simply let $z_t$ be the trajectory starting from the trajectory that achieves the maximum Lyapunov exponent. Again, this dynamics escapes with probability 1 by the Furstenberg-Kesten theorem. The proof is complete. $\square$

### C.2.4 PROOF OF PROPOSITION 3

*Proof.* We consider the dynamics of SGD around a saddle:

$$\ell = -\chi \sum_i u_i w_i, \tag{54}$$

where we have combined $\frac{1}{S}\sum_{(x,y)\in B} xy$ into a single variable $\chi$. The dynamics of SGD is

$$\begin{cases} w_{i,t+1} = w_{i,t} + \lambda\chi u_{i,t}; \\ u_{i,t+1} = u_{i,t} + \lambda\chi w_{i,t}. \end{cases} \tag{55}$$

Namely, we obtain a set of coupled stochastic difference equations. Since the dynamics is the same for all values of the index $i$, we omit $i$ from now on. This dynamics can be decoupled if we consider two transformed parameters: $h_t = w_t + u_t$ and $m_t = w_t - u_t$. The dynamics for these two variables is given by

$$\begin{cases} h_{t+1} = h_t + \lambda\chi h_t; \\ m_{t+1} = m_t - \lambda\chi m_t. \end{cases} \tag{56}$$

We have thus obtained two decoupled linear dynamics that take the same form as that in Theorem 1. Therefore, as immediate corollaries, we know that $h$ converges to 0 if and only if $\mathbb{E}_B[\log|1 + \lambda\chi|] < 0$, and $m$ converges to 0 if and only if $\mathbb{E}_B[\log|1 - \lambda\chi|] < 0$.

When both $h$ and $m$ converge to zero in probability, we have that both $w$ and $u$ converge to zero in probability. For the data distribution under consideration in section 4 and for batchsize one, we have

$$\mathbb{E}[\log|1 + \lambda\chi|] = \frac{1}{2}\log|(1 + \lambda)(1 + \lambda a)| \tag{57}$$

and

$$\mathbb{E}[\log|1 - \lambda\chi|] = \frac{1}{2}\log|(1 - \lambda)(1 - \lambda a)|. \tag{58}$$

There are four cases: (1) both conditions are satisfied; (2) one of the two is satisfied; (3) neither is satisfied. These correspond to four different phases of SGD around this saddle. $\square$

