# OpenReview forum: "Probabilistic Stability of Stochastic Gradient Descent"
_ICLR.cc/2024/Conference — Submitted to ICLR 2024_

### Official Review · Reviewer_PopN · 2023-10-24

**Soundness:** 2 fair
**Presentation:** 3 good
**Contribution:** 2 fair
**Rating:** 5
**Confidence:** 3

**Summary:**

This paper proposes to study the stability of SGD from a probabilistic viewpoint. The argument is that the existing stability analysis based on the convergence as measured by the moment is not sufficient to explain the dynamic behavior of SGD. The main results show that the probabilistic stability of SGD in high-dimension is equivalent to a condition on the sign of the Lyapunov exponent of the SGD dynamics. The derived results provide a new perspective to understand why SGD selects solutions with good generalization from an enormous number of possible solutions.

**Strengths:**

The paper proposes a different perspective to understand the dynamic behavior of SGD, which differs from the existing explanation that SGD prefers flatter solutions. The paper proposes probabilistic stability as a weaker condition than the moment stability. Furthermore, the connection of the probabilistic stability to the Lynapunov exponents is interesting.

**Weaknesses:**

It seems the analysis is not quite rigorous, and I found several gaps in the theoretical analysis.

The Lyapunov exponent seems to be a conservative parameter as it needs to take the maximum over all initializations. It seems that this quantity cannot fully illustrate the behavior of SGD since the initialization also has a large impact on the behavior of the algorithm, which cannot be explained by the Lyapunov exponent.

**Questions:**

Above Eq (16), the paper shows that algorithmic stability holds if $\lambda=1/x_i^2$. Then, Eq (16) says that the largest stable learning rate is $1/x_{min}^2$. Should the largest stable learning rate be $1/x_{max}^2$ since we need the algorithm to be stable for any chosen example, and therefore should choose the smallest step size?

Above Eq (37), the paper shows that $m=tE_x[\log|1-\lambda h_t|]$. However, this identity only holds if $h_1=h_2=\ldots=h_t$. In this case, the matrix $\hat{H}(x)$ should remain the same over the optimization process. This seems to be a strong requirement.

I cannot see how the argument below Eq (39) holds. That is, how to get the convergence by the law of large numbers?

In Eq (46), the paper uses the identity $E[\hat{H}_t\theta_t]=E[\hat{H}_t]E[\theta_t]$. This identity holds if $\hat{H}_t$ and $\theta_t$ are independent. This also seems to be a restrictive condition.

Theorem 3 requires $X_i$ to be independent random matrices. The paper applies Theorem 3 to get Eq (54), which requires $\hat{H}_i$ to be independent. However, it seems that these matrices are not independent, and therefore the Theorem 3 cannot be applied?

I cannot see how Eq (52) holds. In particular, how this identity holds with a small O notation.

Typos:
Eq (6): $\lambda^3$ should be $O(\lambda^3)$
Below Eq (10): ", This" should be ", this"
Proposition 2: "but $L_p$ stable" should be ""but not $L_p$ stable""?

---

> ### Author Response · Authors · 2023-11-22
> **Reply part 1**
>
> Thank you for the detailed and constructive feedback. We answer both the weaknesses and questions below.
>
>
> Weaknesses:
>
> **1. It seems the analysis is not quite rigorous, and I found several gaps in the theoretical analysis.**
>
> Thanks for the criticism. We address the concerns below and have clarified these points in the revision. In short, we believe that most of the criticisms are due to misunderstanding of the definitions or of the problem setting.
>
> **2. The Lyapunov exponent seems to be a conservative parameter as it needs to take the maximum over all initializations. It seems that this quantity cannot fully illustrate the behavior of SGD since the initialization also has a large impact on the behavior of the algorithm, which cannot be explained by the Lyapunov exponent.**
>
> Thank you for the criticism. The Lyapunov exponent is actually more versatile than what we have focused on studying in the manuscript, and it is easy to extend the framework to study the effect of initialization. What we have called the "Lyapunov exponent" is more technically called the "largest Lyapunov exponent." More broadly, the Lyapunov exponent is defined with respect to a specific initialization. If the dynamical variable is $d$-dimensional, a famous result proves that the Lyapunov exponent takes on at most $d$ distinctive values. Therefore, an important future direction is to study the effect of initialization on the Lyapunov exponent and this is not a limitation of the proposed theoretical framework. Also, see footnote 5 in our manuscript for this point.
>
>
> Questions:
>
> **1. Above Eq (16), the paper shows that algorithmic stability holds if $\lambda = 1/x_i^2$. Then, Eq (16) says that the largest stable learning rate is $1/x_\text{min}^2$. Should the largest stable learning rate be $1/x_\text{max}^2$ since we need the algorithm to be stable for any chosen example, and therefore should choose the smallest step size?**
>
> Thanks for this question. While this point might be surprising to some readers, we point out that, it holds true that for any $i$, if $\lambda = 1/x_i^2$, then the algorithm is globally stable, meaning that running SGD converges to the global minimum in probability.
>
> When $\lambda=1/x_{min}$, it might appear unlikely that SGD is stable because it is unstable for the rest of the data points. However, in this very intriguing example, SGD being stable only for a single sample is sufficient to prevent its divergence and guarantee a probabilistic convergence to the global minimum (and this demonstrates the power of the probabilistic stability).
>
>
>
> **2. Above Eq (37), the paper shows that $m=tE_x[\log|1-\lambda h_t|]$. However, this identity only holds if $h_1=h_2=\ldots=h_t$. In this case, the matrix $\hat{H}(x)$ should remain the same over the optimization process. This seems to be a strong requirement.**
>
> Thank you for the comment. This is a misunderstanding. $m$ is an auxiliary constant, which we defined as the expected value. Therefore, there is no requirement or assumption here. It is just a definition we used to simplify the proof. The purpose of introducing $m$ is to define $z_t$ as a zero-mean random variable.
>
>
> **3. I cannot see how the argument below Eq (39) holds. That is, how to get the convergence by the law of large numbers?**
>
> It follows from the fact that $z_t$ is the sum of independent random variables, and so $z_t / t$ is the sample average of the random variable $\log|1 - \lambda h_t| - \mathbb{E}[\log|1 - \lambda h_t|]$. This random variable has zero mean and is identically distributed across $t$. As we always approximate $h$ using the Hessian at the stationary point, there is no dependence of $h$ on $\theta_t$. As a consequence, $h_t$ are independent and so does $\log|1 - \lambda h_t| - \mathbb{E}[\log|1 - \lambda h_t|$. Thus, the law of large numbers applies. The RHS can be obtained taking the limit of $t\to\infty$, and no law of large numbers is required.

---

> > ### Author Response · Authors · 2023-11-22
> > **Reply part 2**
> >
> > **4. In Eq (46), the paper uses the identity $E[\hat{H}_t\theta_t]=E[\hat{H}_t]E[\theta_t]$ . This identity holds if $\hat{H}_t$ and $\theta_t$ are independent. This also seems to be a restrictive condition.**
> >
> > This is not a restrictive condition. Within the framework of linear stability, the dynamics only depends linearly on $\theta$. Therefore, in linear stability, $\hat{H}_t$ is never dependent on $\theta_t$. This assumption of linear stability holds when the model parameters are sufficiently close to a stationary point $\theta^*$, and so it is equivalent to assuming that $ \hat{H}_t(\theta)\approx \hat{H}_t(\theta^*)$ for a constant (or approximately constant) $\theta^*$.
> >
> > That being said, the only way for $\hat{H}_t$ and $\theta_t$ to be dependent is that the random sampling of data points is dependent -- which cannot happen by the standard definition of minibatch sampling. Namely, under SGD, $\theta_t$ depends on $\{h_s, s < t\}$. Therefore, as $h_t$ are i.i.d. as explained in the answer to the previous question, there is no dependence between $\theta_t$ and $h_t$.
> >
> > As an example, please see the problem we studied in proposition 3. It is clear that the $h_t$ matrices are independent in this case. We have clarified this point in the manuscript.
> >
> > **5. Theorem 3 requires $X_i$ to be independent random matrices. The paper applies Theorem 3 to get Eq (54), which requires $\hat{H}_i$ to be independent. However, it seems that these matrices are not independent, and therefore the Theorem 3 cannot be applied?**
> >
> > Please first see our answers above. As explained in the answer to Question 4, the Hessian only depends on the sampled data points which are independent.
> >
> > **6. I cannot see how Eq (52) holds. In particular, how this identity holds with a small O notation.**
> >
> > Eq (52) works in the same way as Eq (39). The little $o$ comes from the fact that $\log \epsilon/t$ vanishes at large $t$. The rest of the proof comes from the independence of sampling data points by the SGD algorithm.
> >
> > **Typos: Eq (6): $\lambda^3$ should be $O(\lambda^3)$ Below Eq (10): ", This" should be ", this" Proposition 2: "but $L_p$ stable" should be ""but not $L_p$ stable""?**
> >
> > Thank you for pointing them out. We have fixed the typos.

---

### Official Review · Reviewer_My7s · 2023-10-28

**Soundness:** 3 good
**Presentation:** 1 poor
**Contribution:** 2 fair
**Rating:** 5
**Confidence:** 3

**Summary:**

This paper introduces probabilistic stability as a new notion for analyzing the dynamics of SGD around critical points.  Specifically, the proposed notion is used to characterize different learning phases of SGD such as correct learning, incorrect learning, convergence to low-rank saddles and unstable learning phase. In particular, the authors provide many insights into the convergence to saddle points based on their probabilistic stability notion.

**Strengths:**

Previous approaches may be insufficient to characterize the dynamics of SGD near saddle points while the proposed probabilistic stability in this paper overcomes this limitation.

**Weaknesses:**

The analysis in this work is restricted to the assumption that initialization point is near a given stationary point. While I acknowledge that such a limitation is not unique to this paper, it deviates from real-world scenarios. Numerous observations have indicated that the early phase of learning substantially impacts the ultimate generalization performance of deep neural networks. Consequently, centering on the dynamics around stationary points may fall short in elucidating the success of deep learning.

In addition, this paper is poorly written and it needs substantial improvement.

**Questions:**

Major concerns:

1. A recent study [1] shows that neural networks trained by gradient-based methods may not necessarily converge to stationary points. In other words, the gradients (norm) may not even vanish when these networks exhibit satisfactory performance. This observation raises questions about the applicability of probabilistic stability in understanding deep learning.

[1] Jingzhao Zhang, et al. "Neural network weights do not converge to stationary points: An invariant measure perspective." ICML 2022.

2. Is the notion of Linear stability as defined in Wu et al. (2018) equivalent to Definition 2 in this paper? Or, is one notion weaker than the other?

3. On Page 3, in the text beneath the caption of Figure 1, the statement "This means that Eq. (1) can be seen as an effective description for only a small subset of all parameters ..." lacks clarity in its flow. Could you provide further elaboration on this?

4. On Page 6, in the fifth-to-last line, you mention, "The theory also shows that if we fix the learning rate and noise level, increasing the batch size makes it more and more difficult to converge to the low-rank solution...". However, in Proposition 3, it's not  clear to me how the batch size explicitly factors into this observation. Could you elaborate more on the role of batch size in this context?

5. Second line on Page 4: "If $\mathbb{E}[h]>0$, the condition is always violated". When $\mathbb{E}[h(x)]>0$, the RHS of Eq.(6) $<0$, doesn't it make the condition in Eq.(5) be valid? Why the condition is violated? Is it a typo or do I misunderstand anything?

Minor comments:

1. Notations are inconsistent in this paper. For example, $n$ serves as the representation of a norm type in the first paragraph on Page 2, where it is a scalar. Later in Section 3.1, $n$ is used as a fixed unit vector. In addition, there is a disparity in the definition of the loss function between Eq.(2) and Eq.(12), as they take different inputs.

2. The paragraph after Definition 1: "A sequence $z_t$ converges" $\Longrightarrow$ "A sequence $\theta_t$ converges"

3. The sentence after Eq.(2): "The dynamics of $w$", and the sentence before Eq.(10): "the following condition implies that $w$...", what is $w$ here? It is not defined.

4. In the capitation of Figure 1: "Phase **I** corresponds to" $\Longrightarrow$ "Phase **Ia** corresponds to"

5. Paragraph after Eq.(7): "The denominator is the Hessian ... signal in the gradient. The denominator is the strength ...", do you mean the numerator for the first "denominator"?

---

> ### Author Response · Authors · 2023-11-22
> **Reply part 1**
>
> Thank you for the detailed and constructive feedback. We answer both the weaknesses and questions below.
>
>
> Weaknesses:
>
> **1. The analysis in this work is restricted to the assumption that initialization point is near a given stationary point. While I acknowledge that such a limitation is not unique to this paper, it deviates from real-world scenarios. Numerous observations have indicated that the early phase of learning substantially impacts the ultimate generalization performance of deep neural networks. Consequently, centering on the dynamics around stationary points may fall short in elucidating the success of deep learning.**
>
> Thanks for this criticism. This is a very important point that we want to clarify. We would like to point out that our theoretical analysis is fully compatible with the observation that the weights do not converge to stationary points but to stationary distributions. We have clarified this point in this revision. See our answer to Question 1 below.
>
>
> **2. In addition, this paper is poorly written and it needs substantial improvement.**
>
> Thanks for this criticism. We have now fully updated the manuscript to incorporate the feedback from all the reviewers.
>
> Questions:
> Major concerns:
>
> **1. A recent study [1] shows that neural networks trained by gradient-based methods may not necessarily converge to stationary points. In other words, the gradients (norm) may not even vanish when these networks exhibit satisfactory performance. This observation raises questions about the applicability of probabilistic stability in understanding deep learning.
> [1] Jingzhao Zhang, et al. "Neural network weights do not converge to stationary points: An invariant measure perspective." ICML 2022.**
>
> Thank you for the criticism and this interesting reference. First of all, we would like to point out that Ref. [1] leaves the possibility open for models to converge to stationary points. For example, Ref. [1] states clearly in section A-1 that “the study of overparametrization and convergenece to stationary point may still be true in many cases.” Thus, the analysis in this paper is likely applicable for the smaller models that are able to overfit the data.
>
> Secondly, our analysis can still apply when the model converges to a distribution, but not to a stationary point. The reason is that it is fully possible (and quite common) for SGD to converge to a point in some subspace, and at the same time converge to a nontrivial distribution overall. As a simple example to illustrate our point, let us consider a linear regression problem trained with SGD, with the per-batch loss: $l(w)=(w^Tx -y)^2$. Suppose $x$ takes a low-rank structure such that with probability one, $x$ is orthogonal to a vector $n$. Then, one can show that with probability one, $n^T \nabla_w l(w) =0$. Namely, in the subspace specified by $n$, the model parameters stay at a single point during training (and thus its series converges to a point in all common notions of stability). In contrast, under common assumptions on the learning rate and data distribution, the dynamics of $w$ in the subspaces orthogonal to $n$ do not converge to a point but to a distribution. Another more advanced example is the phenomenon of the loss of plasticity, common in continual learning. Here, more and more neurons become dead (incoming and outgoing weights becoming zero gradients once and for all) as the training proceeds. In these tasks, the model never converges to fixed points, yet a subset of the parameters do converge to stationary points (for example, see https://arxiv.org/abs/2303.07507).
>
> The reason for this is that the gradient covariance matrix of SGD in deep learning is highly structured and contains many degenerate directions. We have now included a numerical example of matrix factorization to show this: the overall gradient norm does not converge to zero, while the projection of the gradient norm in a subspace does converge to zero -- therefore, in this given subspace, SGD has converged to a stationary point. See Section A5 for the experiment. We have also added these references and discussion to the main text.

---

> > ### Author Response · Authors · 2023-11-22
> > **Reply part 2**
> >
> > **2. Is the notion of Linear stability as defined in Wu et al. (2018) equivalent to Definition 2 in this paper? Or, is one notion weaker than the other?**
> >
> > They only differ in some corner cases and are essentially the same for what we need, which is to prove proposition 2. Using both the definition of Wu et al. (2018) and our version of definition 2, proposition 2 can be proved using the same proof. The point is to show that moment-based stability notions must diverge on a saddle.
> >
> > Since they are not essentially different, we use the current definition of Def 2 because its link to probabilistic stability is well-known (one bounds the other), and it is easier to understand and compare this definition with probabilistic stability. Also, we would also like to point out that the definition of linear stability for SGD is not a textbook result. For example, Mulayoff et al. (2021), one of the references we added for definition 2, defines the linear stability in a slightly different way than in Wu et al. (2018). We have clarified this point in the revision.
> >
> > **3. On Page 3, in the text beneath the caption of Figure 1, the statement "This means that Eq. (1) can be seen as an effective description for only a small subset of all parameters ..." lacks clarity in its flow. Could you provide further elaboration on this?**
> >
> > Thanks for this question. Please first see our answer to question 1. Here, the discussion refers to the fact that the dynamics of SGD in many subspaces are quite independent of the dynamics of the rest of the parameters so one can independently study the dynamics in this subspace.
> >
> > **4. On Page 6, in the fifth-to-last line, you mention, "The theory also shows that if we fix the learning rate and noise level, increasing the batch size makes it more and more difficult to converge to the low-rank solution...". However, in Proposition 3, it's not clear to me how the batch size explicitly factors into this observation. Could you elaborate more on the role of batch size in this context?**
> >
> > Thanks for asking for clarification. The simplest way to see this is to note that using a large batch size leads to smaller gradient noise -- so increasing the batch size leads to a small variance in the gradient. Our original manuscript was implicit about this point. We have now updated proposition 3 and its related discussion to make the dependence on batch size explicit.
> >
> > **5. Second line on Page 4: "If $\mathbb{E}[h] >0$, the condition is always violated". When $\mathbb{E}[h] >0$, the RHS of Eq.(6) < 0, doesn't it make the condition in Eq.(5) be valid? Why the condition is violated? Is it a typo or do I misunderstand anything?**
> >
> > This is a typo. It should be $\mathbb{E}[h] <0$. We have fixed this in the revision.
> >
> > Minor comments:
> >
> > **1. Notations are inconsistent in this paper. For example, $n$ serves as the representation of a norm type in the first paragraph on Page 2, where it is a scalar. Later in Section 3.1, $n$ is used as a fixed unit vector. In addition, there is a disparity in the definition of the loss function between Eq.(2) and Eq.(12), as they take different inputs.**
> >
> > Thank you for the comment. We now use p instead of n for the norm. And, we now use semicolons to separate the dependence of parameters and data in the loss function so that Eq.(2) and Eq.(12) can be written in a consistent way.
> >
> > **2. The paragraph after Definition 1: "A sequence $z_t$ converges" $\to$  "A sequence $\theta_t$ converges"**
> >
> > This is now fixed.
> >
> > **3. The sentence after Eq.(2): "The dynamics of $w$", and the sentence before Eq.(10): "the following condition implies that $w$...", what is $w$ here? It is not defined.**
> >
> > We have fixed this typo. They are supposed to be $\theta$.
> >
> > **4. In the capitation of Figure 1: "Phase I corresponds to" $\to$ "Phase Ia corresponds to"**
> >
> > Thank you for pointing this out. However, we mean to write $I$ here, as in both phase $Ia$ and $Ib$ the model converges in probability to the correct solution (because Ia converges both in prob. and in variance). However, there is a typo in the sentence after this one, and we believe that this typo has created misunderstanding. We have changed “In phase Ib, the model also converges in variance.” to “In phase Ia, the model also converges in variance.”
> >
> > **5. Paragraph after Eq.(7): "The denominator is the Hessian ... signal in the gradient. The denominator is the strength ...", do you mean the numerator for the first "denominator"?**
> >
> > Yes. Thank you for pointing it out.

---

### Official Review · Reviewer_bHzM · 2023-10-31

**Soundness:** 4 excellent
**Presentation:** 2 fair
**Contribution:** 3 good
**Rating:** 5
**Confidence:** 4

**Summary:**

The authors investigate the behavior of SGD through the lens of probabilistic stability: convergence in probability to certain critical points. Dynamics are considered for quadratic losses (justified through local approximation about a critical point), with particular attention given to the effect of saddle points. Necessary and sufficient conditions are provided for probabilistic stability, first for rank-1 quadratic forms, and then for the general case. Probabilistic stability is compared with norm-based stability to highlight the inadequacy of norm-based approaches. Two synthetic examples are considered, and phase diagrams are presented to illustrate different behaviors according to the scale of the noise, and the step size. The phases are delineated as:

- Ia. Probabilistic stability to correct solution
- Ib. Stability in norm to correct solution
- II. Probabilistic stability to incorrect solution
- III. Convergence to low-rank saddle point
- IV. Completely unstable

Universal behavior is implied when experiments on ResNet18 with CIFAR10 also display similar phases. Finally, selection of solutions is considered for SGD with a two-layer network w/ Swish activation, highlighting once again that norm-based convergence is an ineffective criterion.

**Strengths:**

- Linearised dynamics are effective for investigating the late stages of training; well-suited for studying stability.
- Convergence in probability is a much better condition to study than norm-based convergence; this is shown theoretically and empirically.
- The examples are relatively simple, but are effective at demonstrating the phase transitions.
- Universality of the phase diagrams even with real examples is fascinating

**Weaknesses:**

- Studying dynamics of SGD using Lyapunov-type criteria is hardly new, so the theoretical contributions here are particularly limited in their novelty. These conditions for ergodicity of random linear recurrence relations (which immediately imply probabilistic stability as stated here) are already well known [1]. Such conditions have been considered for stochastic optimizers in the ML literature as well [2,3].
- The presentation of the phase diagrams is less than ideal, especially since this is perhaps the key contribution of the paper. Figures are confusing as presented: at the very least, they should be closer to where they are described in text, and empirical results should be compared more directly with theoretical findings. Terms are introduced here that do not appear to be explained.
- Minor: a few typos throughout, e.g. eqn 6 missing an O in front of the cubic term and under eqn 10.

[1] Diaconis, P., & Freedman, D. (1999). Iterated random functions. SIAM Review, 41(1), 45-76.

[2] Gurbuzbalaban, M., Simsekli, U., & Zhu, L. (2021). The heavy-tail phenomenon in SGD. In International Conference on Machine Learning (pp. 3964-3975). PMLR.

[3] Hodgkinson, L., & Mahoney, M. (2021). Multiplicative noise and heavy tails in stochastic optimization. In International Conference on Machine Learning (pp. 4262-4274). PMLR.

**Questions:**

- Is sparsity/density the fraction of zero/non-zero elements in the matrix? This is my best guess, but it is surprising that there would be so many "zero" elements here; is there some larger cutoff which determines whether an element is "zero", or is something else considered here?
- Is Figure 1 comparable to any other empirical examples? If not, what is the purpose of this figure?
- Am I to interpret Figures 4 (right) and 5 (right) as displaying the same behavior as Figure 2 (right)? Can the theoretical prediction be overlaid here too?
- Why is there an arrow from A to B in Figure 6? The text suggests that initialized at B, SGD jumps to C.
- Is Figure 7 converging to Figure 2 as $N \to \infty$?
- I assume convergence to the low-rank saddle should correlate with poor performance?
- Can you outline the phases in the main text? These need to be displayed front and center. It is very confusing to have to refer to a figure legend towards the top of the paper for this.
- Can you put the SGD solution selection part (including Figure 6) near Section 3.2? Otherwise, the purpose of this section is lost on first read.

---

> ### Author Response · Authors · 2023-11-22
> **Reply part 1**
>
> Thank you for the detailed and constructive feedback. We answer both the weaknesses and questions below.
>
>
> Weaknesses:
>
> **1. Studying dynamics of SGD using Lyapunov-type criteria is hardly new, so the theoretical contributions here are particularly limited in their novelty. These conditions for ergodicity of random linear recurrence relations (which immediately imply probabilistic stability as stated here) are already well known [1]. Such conditions have been considered for stochastic optimizers in the ML literature as well [2,3].**
>
> Thank you for pointing this out and for providing the references. We believe this criticism is a misunderstanding of the contribution of our work. First of all, let us clarify that we did not claim to be the first to use the Lyapunov-type criteria to study SGD. We have updated the manuscript to clarify this point.
>
> Our main contribution is to study the stability of saddle points in a neural network landscape -- although the analysis can also be extended to study local minima, this is not the emphasis of our work and so its discussion is only presented in the manuscript. The most important contribution of our work is to show both empirically and theoretically that it is both common and possible for SGD to converge to constrained saddle points -- and a crucial theoretical insight is that this type of convergence can ONLY be studied within the framework of probabilistic stability. We would like to emphasize the second point more: we did not just apply probabilistic stability or Lyapunov-type analysis to these saddles, but we also proved that this is the only correct notion, which is established by proposition 2. Neither of these contributions appear in [2,3] and do not follow trivially from any result in [1].
>
> That being said, these references do relate to ours. However, compared to [2] and [3] and as explained above, our work has a different context and emphasis. Both [2] and [3] focus on studying the dynamics of SGD and its ergodicity near a local minimum. In contrast, the focus of our work is the attractivity and/or repulsiveness of the saddles. Thus, the findings in these papers do not apply immediately to the case we are interested in, that is, the linear stability near the saddle points.
>
> We have cited these works and added discussion regarding their relation to our work.
>
> [1] Diaconis, P., & Freedman, D. (1999). Iterated random functions. SIAM Review, 41(1), 45-76.
>
> [2] Gurbuzbalaban, M., Simsekli, U., & Zhu, L. (2021). The heavy-tail phenomenon in SGD. In International Conference on Machine Learning (pp. 3964-3975). PMLR.
>
> [3] Hodgkinson, L., & Mahoney, M. (2021). Multiplicative noise and heavy tails in stochastic optimization. In International Conference on Machine Learning (pp. 4262-4274). PMLR.
>
>
> **2. The presentation of the phase diagrams is less than ideal, especially since this is perhaps the key contribution of the paper. Figures are confusing as presented: at the very least, they should be closer to where they are described in text, and empirical results should be compared more directly with theoretical findings. Terms are introduced here that do not appear to be explained.**
>
> Thanks for this comment. We have improved the readability of the figures and clarified their connections to the main text. Also, see our answers to the questions below.
>
> That being said, while the phase diagrams are indeed a key contribution of our work, we stress that it is not the only crucial contribution we have. As we pointed out in the answer to Weakness 1, the first (and perhaps more important) contribution of our work is to show that the saddle points in deep learning have to be studied within the framework of probabilistic stability -- here, proposition 2 plays a key role as it proves that no other existing stability definition can be used to analyze the attractivity of these saddles.
>
>
> **3. Minor: a few typos throughout, e.g. eqn 6 missing an O in front of the cubic term and under eqn 10.**
>
> Thanks for pointing this out. We have fixed this typo in the revision.

---

> > ### Author Response · Authors · 2023-11-22
> > **Reply part 2**
> >
> > Questions:
> >
> > **1. Is sparsity/density the fraction of zero/non-zero elements in the matrix? This is my best guess, but it is surprising that there would be so many "zero" elements here; is there some larger cutoff which determines whether an element is "zero", or is something else considered here?**
> >
> > Thanks for this question. The sparsity is indeed the fraction of zero elements in the matrix. First of all, we note that we do observe exact zeros in the experiments -- but this often takes quite long to achieve. More often, we set a threshold, and any quantity smaller than the threshold is considered zero. Typically, the threshold is set to be 1e-8, which is roughly the machine precision level of Pytorch.
> >
> > **2. Is Figure 1 comparable to any other empirical examples? If not, what is the purpose of this figure?**
> >
> > Figure 1-left is an example of Proposition 3 (when N=2) studied in Section 4. The phase boundaries are drawn according to proposition 3 and its experimental details are given in A.1. Figure 1-left is the case when there are only two data points for proposition 3. For a larger N, the results are given in Figure 7. For infinite N, the result is shown in Figure 2. This has been clarified in the updated Section 4.
> >
> > Also, Figure 1-right is more of an illustration of how the phase boundaries are decided by the Lyapunov exponent.
> >
> >
> > **3. Am I to interpret Figures 4 (right) and 5 (right) as displaying the same behavior as Figure 2 (right)? Can the theoretical prediction be overlaid here too?**
> >
> > Figure 4 and 5 indeed displays the same qualitative behavior as Figure 2. However, the order of magnitude for the learning rate in the figures is different.  Figure 3 is the multi-dimensional version of Figure 2, and the theory curve in Figure 3 corresponds to the dashed curve in Figure 2.
> >
> > Figure 4 and Figure 5, obtained using realistic data and ResNet, demonstrate the same qualitative behavior as Figure 3. However, we do not have a quantitative prediction for the precise phase boundary for ResNet because it is not tractable to compute its per-batch Hessians. In summary, the agreement between Figure 2 and 3 is quantitative, while the agreement between Figure 2 and Figures 4-5 is qualitative.
> >
> > **4. Why is there an arrow from A to B in Figure 6? The text suggests that initialized at B, SGD jumps to C.**
> >
> > The red arrow from A to B is the prediction by L2 stability: when the learning rate is large, even if initialized at A, SGD will be unstable there and must jump to an alternative solution -- and this certainly is not what happens in reality. The text referring to this is located in the middle paragraph in section 5. We have added a description for this in the caption of Figure 6 to avoid misunderstanding.
> >
> > **5. Is Figure 7 converging to Figure 2 as $N \to \infty$.?**
> >
> > Yes.
> >
> > **6. I assume convergence to the low-rank saddle should correlate with poor performance?**
> >
> > Not necessarily. At its face value, low-rankness means only that the capacity of the model is limited. This could be a bad thing when a high model capacity is desired. This could be good if overfitting is a concern. Both perspectives appear frequently in the context of machine learning and deep learning.
> >
> > **7. Can you outline the phases in the main text? These need to be displayed front and center. It is very confusing to have to refer to a figure legend towards the top of the paper for this.**
> >
> > Thanks for this suggestion. We have updated Section 4 to clearly outline the phases.
> >
> > **8. Can you put the SGD solution selection part (including Figure 6) near Section 3.2? Otherwise, the purpose of this section is lost on first read.**
> >
> > Thank you for the proposal. However, we prefer to separate the theory and its implications, so the organization of the contents remains roughly the same. To help readers who want to see an immediate example after Section 3.2, we have added a reference to section 5 as well as Fig. 6 in section 3.2 so that the purpose of 3.2 is not lost.

---

### Official Review · Reviewer_Mfzw · 2023-11-01

**Soundness:** 2 fair
**Presentation:** 2 fair
**Contribution:** 2 fair
**Rating:** 3
**Confidence:** 3

**Summary:**

The paper proposes a new stability notion, i.e., probabilistic stability, to study the stability of the SGD learning algorithm. The goal of proposing the new stability notion is to explain why deep learning models trained with SGD generalize well. The paper also revisits some variance-based stability notions and illustrates that those stability notions cannot explain the convergence of SGD.

**Strengths:**

1. The problem studied in this paper, i.e., explaining deep learning phenomena using a new stability notion, is valuable and interesting.
2. The paper tackles this problem from a different angle, such as characterizing SGD dynamics from control theory.

**Weaknesses:**

1. Literature on the variance-based stability notion is not adequately discussed in this paper. The paper presents the definition of the variance-based stability notion in Definition 2, but it is unclear what the current results are regarding this type of stability. Additionally, the reference for this stability notion cannot be found in this paper. It would be beneficial to include more discussions regarding the related work.
2. The clarity of the paper can be improved. It is difficult to follow and extract the key points of each section that the paper wants to deliver. It is also hard to connect each section. For example, Section 3.1 shows that rank-1 dynamics are solvable, but how this connects with probabilistic stability is unclear. Section 3.2 jumps to the point that variance-based stability is insufficient, and it is hard to connect these two sections.
3. The technical soundness and significance can be improved. Section 3 show that the linearized dynamics of SGD converges with probability under certain conditions, but it is difficult to establish a connection between this result and how it explains the generalization of SGD in deep learning. Section 4 discusses different phases of SGD learning, but it is unclear how these phases relate to the stability notions proposed in the paper and how they explain the generalization of SGD.

**Questions:**

1. Can we understand that the probabilistic stability defined in this paper is more like a convergence guarantee?
2. What is the literature on variance-based stability?
3. In Definition 1 and 2, it would be great to clarify whether θ* is fixed or a random variable. After Definition 1, there is a typo in the convergence in probability, "< ε" should be revised to "> ε."
4. Some typos:
    1. After equation (2): "The dynamics of w thus obeys Eq. (1)." Here, w is not defined.
    2. Before equation (10): "Then, the following condition implies that w → p 0:". Also, w is not defined.


5. Figure 1: It would be great to explain the x-axis and y-axis in the caption (the same in Figure 2). Also, what are w_t and u_t? How are the phases (different colors) calculated based on values on the x-axis and y-axis?
6. Figure 3: It would be great to explain how the color map is calculated and how convergence is calculated.

---

> ### Author Response · Authors · 2023-11-22
> **Reply part 1**
>
> Thank you for the detailed and constructive feedback. We answer both the weaknesses and questions below.
>
>
> Weaknesses:
>
> **1. Literature on the variance-based stability notion is not adequately discussed in this paper. The paper presents the definition of the variance-based stability notion in Definition 2, but it is unclear what the current results are regarding this type of stability. Additionally, the reference for this stability notion cannot be found in this paper. It would be beneficial to include more discussions regarding the related work.**
>
> Thank you for the detailed comments. We believe that your main misunderstanding comes from our insufficient discussion of why the notion of stability is important.
>
> In the context of deep learning, the main purpose for both the variance-based stability (previous works) and probabilistic stability (this work) is to understand which class of solutions or stationary points of the loss function is preferred by the SGD-based training. The most well-known result of the variance-based stability is that at a large learning rate or small batch-size, SGD prefers flat local minima because sharp minima are unstable, and, thus, repulsive, and so SGD will eventually escape sharp minima because of unstable oscillation and fluctuation (for example, see Wu et al. (2018)). However, a major limitation of the variance-based stability is that it cannot be applied to understand the stability of saddle points -- which is the main topic of our work.
>
> We have added references to the variance-based stability and clarified these points. See the discussion below definition 2.
>
>
>
> **2. The clarity of the paper can be improved. It is difficult to follow and extract the key points of each section that the paper wants to deliver. It is also hard to connect each section. For example, Section 3.1 shows that rank-1 dynamics are solvable, but how this connects with probabilistic stability is unclear. Section 3.2 jumps to the point that variance-based stability is insufficient, and it is hard to connect these two sections.**
>
> Thank you for the comment. Please first see our answer to weakness 1 and note that the main focus of our work is this: why and when are saddle points attractive under SGD. It is to serve this purpose that the manuscript is organized.
>
> While there has been various empirical evidence that saddle points are attractive, there has not been any theory to show and understand how and why SGD can converge to these saddle points in deep learning. Our goal is to fill this important theoretical gap. Section 3 solves rank-1 dynamics under the probabilistic stability -- the main message here is that saddle points can indeed become attractive under the probabilistic stability under many choices of hyperparameter (large learning rate, for example).
>
> The next question is whether we have any other theoretical tool to understand the attractivity of saddles under SGD, and this is the purpose of Section 3.2. Section 3.2 shows that perhaps surprisingly, no notions of stability based on the statistical moments can help us understand the attractivity of saddles. This establishes the probabilistic stability as something quite unique -- because one must study the probabilistic stability to understand the attractivity of the saddles.
>
> In summary, sections 3.1 and 3.2 together establish that probabilistic stability is especially suited and perhaps the ONLY tool for studying why the saddle points are attractive in deep learning.

---

> > ### Author Response · Authors · 2023-11-22
> > **Reply part 2**
> >
> > **3. The technical soundness and significance can be improved. Section 3 show that the linearized dynamics of SGD converges with probability under certain conditions, but it is difficult to establish a connection between this result and how it explains the generalization of SGD in deep learning. Section 4 discusses different phases of SGD learning, but it is unclear how these phases relate to the stability notions proposed in the paper and how they explain the generalization of SGD.**
> >
> >
> > Thanks for this important question. To explain and analyze the practical good generalization of SGD, two crucial questions need to be answered in order: which solution does SGD prefer; what is the nature of these solutions (in terms of its generalization power). The main focus of our analysis is on the first question. Our key contribution here is to show that saddle points that feature special, constrained solutions are preferred when the learning rate is large or the gradient noise is strong.
> >
> > In contrast, the second question is only answered in a qualitative manner in our work. Namely, we show that the learning can be divided into four different phases, and the generalization ability of these four phases is clear -- in the trivial phase, the model generalizes like a trivial model that only outputs a constant; in the correct learning phase, the model generalizes like a model that has reached the global minimum (to which the conventional analysis that links the training error and generalization error can be applied). In the incorrect phase, the model learns the spurious correlations and overfits to noises in the label.
> >
> > How are these phases related to probabilistic stability? In the example of Section 4, the trivial phase is when both $u-w$ and $u+w$ are probabilistically stable at $0$. The correct learning phase is when $u-w$ is prob. stable but $u+w$ is p-unstable (at zero). The incorrect phase is when $u+w$ is prob. stable and $u-w$ is p-unstable. The unstable phase is when both quantities are p-unstable. We have made this point clear in the revision (see Section 4).
> >
> > That being said, the full quantitative analysis of generalization is, however, beyond the scope of this work.
> >
> > Questions:
> >
> > **1. Can we understand that the probabilistic stability defined in this paper is more like a convergence guarantee?**
> >
> > Not quite. A stable solution is an attractive solution. At the end of training, SGD can only stay close (and sometimes converge) to a solution that is stable. It can also be seen as a convergence guarantee IF one is initialized close enough to such a solution -- that being that a solution is stable does not imply that starting from an arbitrary initialization, one will converge to such a solution. For example, for a small learning rate, all global minima are stable, but SGD is more likely to converge to a minimum that is close to its initialization.
> >
> > **2. What is the literature on variance-based stability?**
> >
> > In the context of deep learning, the main purpose for both the variance-based stability (previous works) is to understand which class of solutions or stationary points of the loss function is preferred by the SGD-based training. The most well-known result of the variance-based stability is that at a large learning rate or small batch-size, SGD prefers flat local minima because sharp minima are unstable, and, thus, repulsive, and so SGD will eventually escape sharp minima because of unstable oscillation and fluctuation (for example, see Wu et al. (2018)). However, a major limitation of the variance-based stability is that it cannot be applied to understand the stability of saddle points -- which is the main topic of our work. Also, see our answers above.
> >
> > There are other works that are based on the variance-based stability, for example, Wu & Su (2023), Ma & Ying (2021), Mulayoff et al. (2021). These works study the implicit regularization of SGD by linking the desired properties of solutions to the linear stability of SGD.
> >
> >
> >
> > __3. In Definition 1 and 2, it would be great to clarify whether $\theta^*$ is fixed or a random variable. After Definition 1, there is a typo in the convergence in probability, "< ε" should be revised to "> ε."__
> >
> > The $\theta^*$ is fixed. It is a stationary point in the loss landscape. Thank you for the observation. We have fixed the typo.

---

> > > ### Author Response · Authors · 2023-11-22
> > > **Reply part 3**
> > >
> > > Typos:
> > >
> > > Thanks for pointing out the typos. We have fixed these problems in the updated manuscript.
> > >
> > > **After equation (2): "The dynamics of w thus obeys Eq. (1)." Here, w is not defined.**
> > >
> > > We have fixed this. It should be $\theta$.
> > >
> > > **Before equation (10): "Then, the following condition implies that w → p 0:". Also, w is not defined.**
> > >
> > > We have fixed this also. It should be $\theta$.
> > >
> > >
> > > **Figure 1: It would be great to explain the x-axis and y-axis in the caption (the same in Figure 2). Also, what are w_t and u_t? How are the phases (different colors) calculated based on values on the x-axis and y-axis?**
> > >
> > > Thank you for the question. We use $\lambda$ as the learning rate and $\Lambda$ as the Lyapunov exponent consistently throughout the manuscript, so we didn’t add further explanations for them in the caption due to page limitation. In Figure 1, we added a reference for parameter $a$. In Figure 2, we added explanations for the parameters $\mu$ and $S$. The phase diagram in Figure 1 is calculated based on the results in C.2. The phases are determined by the signs of equations (59) and (60). Please refer to the reply to the third point in Weaknesses for a detailed explanation of how to connect the convergence of $u + w$ and $u-w$ to the phases.
> > >
> > > **Figure 3: It would be great to explain how the color map is calculated and how convergence is calculated.**
> > >
> > > The color map denotes the number of non-zero values of the singular value of matrix $u$ using the model specified in section 4, equation(12). We say the experiment converges if we observe plateaux in the rank-time plots. We have updated Figure 3 to clarify this.

---

### Author Response · Authors · 2023-11-22
**Rebuttal summary**

We would like to thank all reviewers for their comments and suggestions. After carefully studying the reviews, we realized that our paper could be improved in terms of clarity. We added missing discussions and references as suggested by the reviewer and, more importantly, emphasized the context and contribution of our work in the introduction.

It appears to us that we didn’t state the context and contribution of this work clearly, and this has led to misunderstandings. In addition to the added discussions in the introduction of the manuscript, we also would like to explain here the context of our work, our motivation, and our contribution. This manuscript studies the notion of linear stability of SGD. The linear stability analysis studies whether the system stays near a stationary point or escapes it. In the context of machine learning, the framework of linear stability is used to study minima selection by the optimization algorithm (and thus linked to generalization), see Wu et al. (2018). That is to say, when the optimization algorithm enters the neighborhood of a stationary point, will it converge to the stationary point or escape it. A main focus of previous related works is to study the local minima of neural networks.

In contrast, our theoretical and empirical study of SGD shows that it converges to not only minima but also saddle points. The first crucial contribution of our work is to show that the current notion of linear stability of SGD does not apply properly to saddle points, and that the probabilistic stability is the only suitable framework to study the attractivity of saddle points (proposition 2). Reviewers Mfzw, bHzM, and PopN seem to misunderstand this part of the contribution. As a consequence, we obtained the phases of learning and insights regarding the minima preferred by SGD. These implications are the second contribution of the manuscript and all the reviewers seem to like this part.

We have extensively updated the discussions in the manuscript to better reflect our contribution and clarify various points raised by the reviewers. The updated text is colored in orange. The following are the main changes to the manuscript:
1. The abstract, introduction, and conclusions are updated to emphasize our contribution towards understanding the attractivity of the saddle points
2. Clarifications are made to differentiate our with previous works that also study Lyapunov-type analysis
3. The five different phases of learning are now clearly outlined and discussed in Section 4
4. The proposition 3 is updated to explicitly state the dependence of the critical learning rate on the batch size
5. An experiment (section A5) on matrix factorization is added to show that our theory is still applicable even if the model converges to a stationary distribution but not to a stationary point
6. Typos and inconsistencies are now removed

We make detailed replies to the feedback of the reviewers below.

---

### Meta-Review · Area_Chair_54V7 · 2023-12-07

**Metareview:**

The work outlines limitations of a variance/moment based approach to understanding stability of SGD and presents an approach based on the notion of probabilistic stability. The implications of such probabilistic stability include convergence to saddle points or to a distribution, some of which became more clear based on the questions from the reviewers and the corresponding responses.

The reviewers mostly appreciate the novel direction being pursued in the paper, but some concerns regarding the development have been shared, ranging from limited clarity of exposition of the original submission to limited novelty of the toolbox/methods, limited scope given the focus on close to stationary points, difficulty in understanding key contributions such as phase diagrams, etc. To some degree, the authors have responded to some of these concerns.

The work is promising and would have arguably gotten more positive response if the original submission had been more polished, with more clarity of exposition.

**Justification For Why Not Higher Score:**

There were several concerns on the original submission, many related to limited clarity of exposition, potentially limited scope, and potential lack of novelty. In spite of the original submission having such issues, the reviews were fairly detailed and has arguably helped polish the manuscript.

**Justification For Why Not Lower Score:**

N/A

---

### Decision · Program_Chairs · 2024-01-16

Reject